# Human tumor suppressor protein Pdcd4 binds at the mRNA entry channel in the 40S small ribosomal subunit

Jailson Brito Querido [1,2,3,4,6] ✉, Masaaki Sokabe [5,6], Irene Díaz-López [1,6], Yuliya Gordiyenko[1], Philipp Zuber[1], Yifei Du [1], Lucas Albacete-Albacete[1], V. Ramakrishnan [1] ✉ & Christopher S. Fraser [5] ✉

Translation is regulated mainly in the initiation step, and its dysregulation is implicated in many human diseases. Several proteins have been found to regulate translational initiation, including Pdcd4 (programmed cell death gene 4). Pdcd4 is a tumor suppressor protein that prevents cell growth, invasion, and metastasis. It is downregulated in most tumor cells, while global translation in the cell is upregulated. To understand the mechanisms underlying translational control by Pdcd4, we used single-particle cryo-electron microscopy to determine the structure of human Pdcd4 bound to 40S small ribosomal subunit, including Pdcd4-40S and Pdcd4-40S-eIF4A-eIF3-eIF1 complexes. The structures reveal the binding site of Pdcd4 at the mRNA entry site in the 40S, where the C-terminal domain (CTD) interacts with eIF4A at the mRNA entry site, while the N-terminal domain (NTD) is inserted into the mRNA channel and decoding site. The structures, together with quantitative binding and in vitro translation assays, shed light on the critical role of the NTD for the recruitment of Pdcd4 to the ribosomal complex and suggest a model whereby Pdcd4 blocks the eIF4F-independent role of eIF4A during recruitment and scanning of the 5′ UTR of mRNA.

Translation regulation is critical for cellular homeostasis, and dysregulation is implicated in many diseases, such as cancer[1–3]. The regulation of translation mostly occurs during the initiation step[2]. This process starts with the formation of a 43S pre-initiation complex (43S), comprising the 40S small ribosomal subunit, eukaryotic initiation factors eIF1, eIF1A, eIF3, eIF5, and a ternary complex (TC) of guanosine 5′-triphosphate (GTP) bound eIF2 and methionine initiator transfer RNA (tRNAi^Met). The assembled 43S is recruited at the 5′-end of mRNA to form a 48S initiation complex (48S), which then scans along the 5′ UTR of mRNA until it encounters and selects a start codon. The recruitment and scanning processes are assisted by the heterotrimeric cap-binding complex eIF4F, consisting of a scaffold protein eIF4G, a

7-methylguanosine (m⁷G) cap-binding protein eIF4E, and a DEAD-box RNA helicase eIF4A.

Given the importance of eIF4A in translation initiation, it is not surprising that this RNA helicase is a target of different translational control strategies[1], including that by Pdcd4 (programmed cell death 4)[4–10]. Pdcd4 is a tumor suppressor protein known to suppress cell growth, tumor invasion, and metastasis, and is downregulated in nearly all solid tumors. Pdcd4 has two N-terminal RNA recognition motifs (RRM1 and RRM2) followed by middle and C-terminal MA3 domains (MA3m and MA3c) homologous to eIF4A binding domains in eIF4G. In contrast to eIF4G, MA3 domains in Pdcd4 are known to bind eIF4A to inhibit its mRNA binding, ATPase, and helicase activities[4–8,11].

[1]MRC Laboratory of Molecular Biology, Cambridge, UK. [2]Department of Biological Chemistry, University of Michigan, Ann Arbor, MI, USA. [3]Life Sciences Institute, University of Michigan, Ann Arbor, MI, USA. [4]Center for RNA Biomedicine, University of Michigan, Ann Arbor, MI, USA. [5]Department of Molecular and Cellular Biology, College of Biological Sciences, University of California, Davis, CA, USA. [6]These authors contributed equally: Jailson Brito Querido, Masaaki Sokabe, Irene Díaz-López. ✉e-mail: jquerido@umich.edu; ramak@mrc-lmb.cam.ac.uk; csfraser@ucdavis.edu

The function of the RRMs for translational inhibition is not known, despite them being required for complete translational inhibition by Pdcd4[12–14]. It is possible that Pdcd4 binds to eIF4A and prevents the assembly of the cap-binding complex eIF4F. However, eIF4A is present in significant excess over Pdcd4 in cells[15], which suggests that the translational control activity of Pdcd4 goes beyond its direct interaction with eIF4A. Alternative models have proposed that Pdcd4 interacts with eIF4A that is bound to eIF4F, thereby reducing the helicase activity of eIF4F[10]. Our recent structure of a human 48S complex revealed a second, eIF4F-independent molecule of eIF4A bound at the mRNA entry site on the opposite side of the 40S from eIF4F that presumably plays a role in unwinding secondary structure in the 5′ UTR[16]. Thus, it remains unknown whether Pdcd4 blocks the helicase activity of the eIF4A protein that is part of the eIF4F complex, or the second molecule of eIF4A at the leading edge of the 40S subunit[2]. In addition, the functional role of the RRMs located in the N-terminal domain (NTD) of Pdcd4 remain unknown. It has been shown that the NTD is required for Pdcd4 to co-migrate with 40S ribosomal complexes in a sucrose gradient[14]. This interaction was speculated to involve the interaction between Pdcd4, PABP, and mRNA. More recently, a formaldehyde cross-linking of Pdcd4 with salt-washed purified 40S subunits followed by a sucrose gradient suggested a possible direct interaction between Pdcd4 and 40S[17]. Nevertheless, it remained unknown whether and how Pdcd4 could directly bind to the 40S and if it targets eIF4A at the mRNA entry site or eIF4F at the mRNA exit site. Here, we determined high-resolution structures of human Pdcd4 bound to the 40S subunit, including the Pdcd4-40S and Pdcd4-40S-eIF4A-eIF3-eIF1 complexes. These structures reveal the mechanism by which the NTD prevents mRNA accommodation to the mRNA entry channel and decoding site of the 40S subunit while the CTD binds to a second copy of eIF4A located at the mRNA entry site rather than eIF4F.

## Results

### Cryo-EM structure of 40S-Pdcd4 complex

To test the possible direct interaction between Pdcd4 and the 40S subunit, we used native gel assay (NGA) to analyze a direct interaction between a fluorescently modified recombinant Pdcd4 (FL-Pdcd4) and salt-washed 40S subunits (Fig. 1A). We see a co-migration of FL-Pdcd4 with the 40S, which demonstrates a direct interaction between them. To understand further the contribution of each domain of Pdcd4 for its interaction with the 40S, we performed the binding assay in the presence of excess unlabeled competitors. While the wild-type Pdcd4 (residues 1-469) and Pdcd4-NTD (1-156) prevent the binding of FL-Pdcd4 to the 40S, the Pdcd4-CTD (157-469) does not compete for the binding (Fig. 1A), which suggests that the NTD is the primary region involved in the direct interaction with the 40S. Consistent with this finding, a fluorescence anisotropy competition assay reveals that Pdcd4-NTD has similar affinity for the 40S ($K_i = 21$ nM) when compared with the full-length protein ($K_i = 23$ nM) (Fig. 1B).

To characterize the interaction between Pdcd4 and 40S subunit in molecular detail, we used single-particle cryo-electron microscopy (cryo-EM) to determine the structure of a reconstituted human Pdcd4-40S complex at an overall resolution of 2.6 Å (Fig. 1C, Table 1, and Supplementary Figs. 1 and 2). Interestingly, the cryo-EM reconstruction reveals density corresponding to the 40S subunit and an additional density in the mRNA channel of the 40S that has not been described before (Fig. 1C, D). This additional density is only present when the 40S is incubated with Pdcd4 (Supplementary Fig. 3). Moreover, the local resolution of 2.2 Å allowed us to assign it to Pdcd4-NTD and build an atomic model for residues 99 to 145 (Fig. 1E and Table 1). The rest of the protein is not visible, presumably because of high flexibility.

Although we also included eIF1 and eIF1A in the cryo-EM sample to prevent 40S dimerization, we did not see any density corresponding to these factors in the resulting structure of Pdcd4-40S complex.

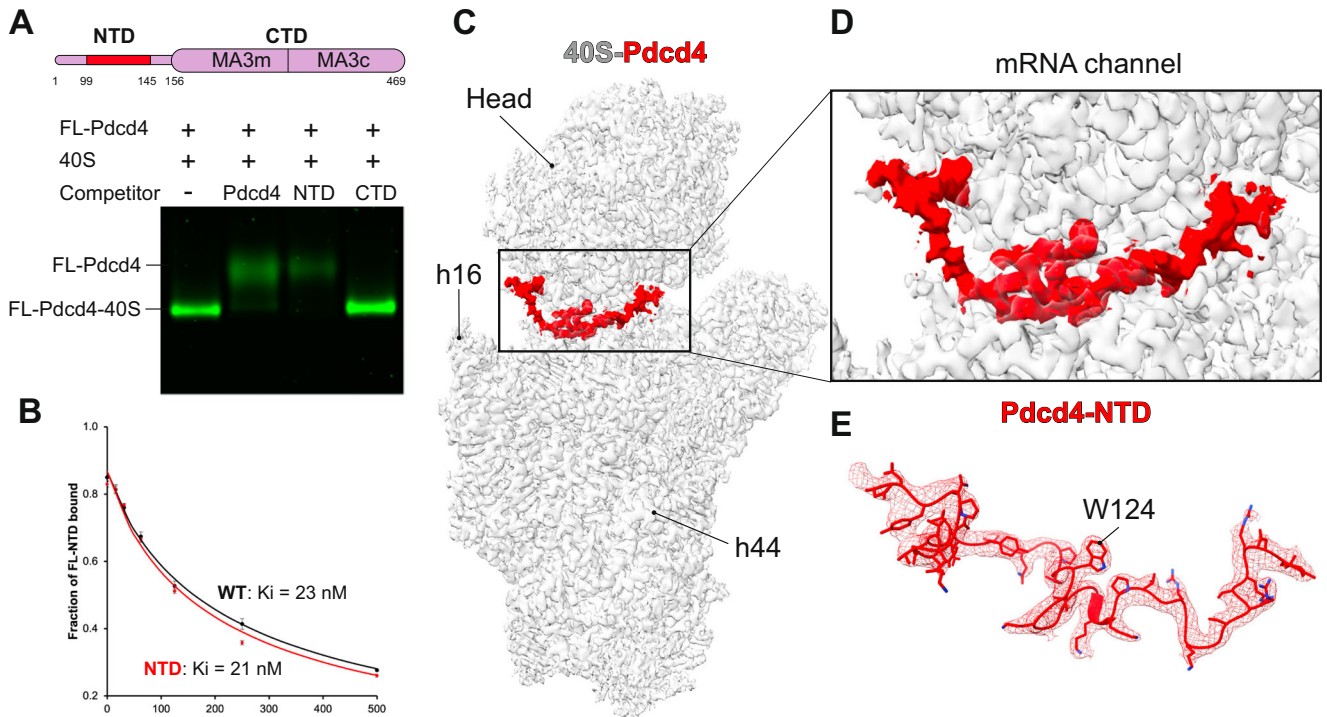

**Fig. 1 | Cryo-EM structure of human Pdcd4 bound to the 40S subunit. A** Native gel assay showing a fluorescence-labeled-Pdcd4 binding to 40S subunit in the absence or presence of excess unlabeled Pdcd4 (residues 1-469) or its fragment (NTD:1-156, or CTD:157-469). **B** Fluorescence anisotropy competition assay showing dissociation of labeled Pdcd4-NTD from 40S upon titration of unlabeled Pdcd4 or Pdcd4-NTD. Error bars represent the mean ± SEM ($n = 3$). For the NTD, only two independent experiments were performed. The experiment was repeated at least three times with similar results. **C, D** Cryo-EM reconstruction of human Pdcd4 bound to the 40S small ribosomal subunit. **E** The structure of Pdcd4-NTD and its corresponding cryo-EM density map shown as mesh.

**Table 1 | Cryo-EM data collection, refinement, and validation statistics**

| | #40S-Pdcd4 (EMDB- 44641) (PDB 9BKD) | #Pdcd4-40S-eIF4A- eIF3-eIF1 (EMDB- 44671) (PDB 9BLN) |
|---|---|---|
| *Data collection and processing* | | |
| Magnification | 105,000 | 105,000 |
| Voltage (kV) | 300 | 300 |
| Electron exposure (e−/Å²) | 1.01 | 0.93 |
| Defocus range (µm) | −1.2 to −3.0 | −1.2 to −3.0 |
| Pixel size (Å) | 0.826 | 0.86 |
| Symmetry imposed | C1 | C1 |
| Initial particle images (no.) | 865,286 | 938,386 |
| Final particle images (no.) | 85,656 | 8,515 |
| Map resolution (Å) | 2.57 | 3.9 |
| FSC threshold | 0.143 | 0.143 |
| Map resolution range (Å) | 2.2–6.2 | 2.9–15 |
| *Refinement* | | |
| Initial model used (PDB code) | 6ZMW | 6ZMW |
| Model resolution (Å) | 2.6 | 3.8 |
| FSC threshold | 0.5 | 0.5 |
| Model resolution range (Å) | 1.9–2.6 | 2.6–3.8 |
| Map sharpening $B$ factor (Å²) | −25 | −10 |
| Model composition | | |
| Non-hydrogen atoms | 74,436 | 104,3289,910 |
| Protein residues | 4829 | 112 |
| Ligands | 111 | – |
| *B factors (Å²)* | | |
| Protein | 55.61 | 89.58 |
| Ligands | 37.66 | 38.59 |
| R.m.s. deviations | | |
| Bond lengths (Å) | 0.009 | 0.007 |
| Bond angles (°) | 0.863 | 1.152 |
| Validation | | |
| MolProbity score | 1.51 | 1.76 |
| Clashscore | 2.31 | 4.16 |
| Poor rotamers (%) | 2.55 | 1.66 |
| Ramachandran plot | | |
| Favored (%) | 96.84 | 94.15 |
| Allowed (%) | 3.16 | 5.45 |
| Disallowed (%) | 0.00 | 0.4 |

## Pdcd4-NTD binds to mRNA channel in the 40S subunit

The contribution of NTD to the translational control activity of Pdcd4 remained unclear. Our near-atomic resolution structure reveals molecular details of its interaction with the 40S subunit. Pdcd4-NTD binds at the mRNA entry site and extends towards the P site of the 40S (Fig. 1C). The structure reveals multiple interactions of Pdcd4 with ribosomal protein uS3 (Fig. 2A) and 18S ribosomal RNA (rRNA) in the mRNA channel. The interaction of Pdcd4 with uS3 occurs through multiple residues (Supplementary Table 1), including that between W124 in Pdcd4 and V115-R116 in uS3 (Fig. 2A and Supplementary

Table 1). The W124 anchors into a deep hydrophobic pocket of uS3 (Fig. 2B), which may have a critical role in the interaction with the 40S. Furthermore, the structure reveals a crucial interaction between K114-K115 in Pdcd4 and h18 of the 18S rRNA at the mRNA entry channel (Fig. 2C). To determine the extent to which these residues are important in stabilizing the binding of Pdcd4 with the 40S subunit, we generated two double mutants of human Pdcd4 – V123A/W124A and K114A/K115A. Our fluorescence anisotropy assay data reveals that both mutations abolish the binding of Pdcd4-NTD to the 40S (Fig. 2D).

## Pdcd4-NTD blocks the decoding center of the 40S subunit

In our structure, we see an unusual 40S head swivel movement beyond the closed conformation (Supplementary Fig. 4A), resulting in even narrower and less accessible mRNA binding cleft. Interestingly, this conformation is essentially identical to that found in the structures of an dormant 80S ribosome in complex with hyaluronan-binding protein 4 (Habp4)[18], mammalian SERBP1 (SERPINE1 mRNA binding protein 1), or its counterpart Stm1 in yeast (Supplementary Fig. 4B)[19–22]. These proteins are suggested to inhibit binding of mRNA and translation factors to the ribosome to preserve it in an inactive state. Indeed, the 40S binding region of Pdcd4-NTD has a high local structural and sequence similarity with a corresponding region in these proteins, especially with SERBP1 (Supplementary Fig. 4B, C)[20,23].

Along with the 40S head swivel, the structure reveals an extensive interaction of Pdcd4-NTD with the mRNA binding cleft, including residues R110 and R102 with universally conserved 18S rRNA decoding bases C1698 and C1701 in the A- and P- sites, respectively (Fig. 3A). This interaction also involves many other 18S rRNA bases both in the 40S head and body (Supplementary Table 1), suggesting that the unique 40S conformation and Pdcd4-NTD binding mutually stabilize each other. Superimposing our structure with a structure of the 48S complex in a closed/$P_{in}$ state shows a significant overlap between Pdcd4-NTD and mRNA across the mRNA binding cleft, most noticeably around the entry channel[24] (Fig. 3B). The binding of Pdcd4-NTD across the mRNA binding cleft suggests that it can sterically prevent mRNA accommodation into the channel, which is verified using a fluorescence anisotropy binding assay showing that Pdcd4-NTD reduces the affinity of a mRNA for the 40S by 4-fold (Fig. 3C). This superposition also suggests a possible steric clash between Pdcd4-NTD and $tRNA_i^{Met}$ in the $P_{in}$ state (Fig. 3D), although we note that this may not occur when $tRNA_i^{Met}$ is in the $P_{out}$ state (Fig. 3E). Similarly, although Pdcd4-NTD shows a minor clash with eIF1A at the A-site when superposed on the structure of the body of 43S in the open conformation, it does not clash when superposed on the head in the same structure (Fig. 3F). Because Pdcd4-NTD interacts extensively with both 40S head and body around the A-site, these models imply that Pdcd4-NTD likely adopts an alternative conformation, presumably with fewer contacts with the 40S, when the 40S is in the open conformation. Overall, the structural comparisons around the A- and P-sites imply that the stable binding of Pdcd4-NTD and the resulting steric hindrance in this region depends on the unique 40S head swivel conformation observed in the current structure. Consistent with this, an anisotropy binding assay using FL-Pdcd4-NTD shows a > 10-fold reduction in its affinity for the 40S in the presence of saturating eIF1 and eIF1A, and >100-fold reduction in its affinity for the 43S (Supplementary Fig. 5), suggesting that Pdcd4-NTD on its own cannot stably bind and induce the 40S head swivel in the 43S.

## Pdcd4-CTD interacts with eIF4A at the mRNA entry site

To further investigate the structural basis of translational control by Pdcd4, we reconstituted a human 48S complex in which we included all the eIFs, a capped mRNA containing a long 5′ UTR (105 nucleotides), an AUG codon followed by a short CDS and a poly(A) tail, along with PABP and Pdcd4, for cryo-EM analysis. Although the majority of particles on a grid are identified to be either the 43 S/48S with no Pdcd4

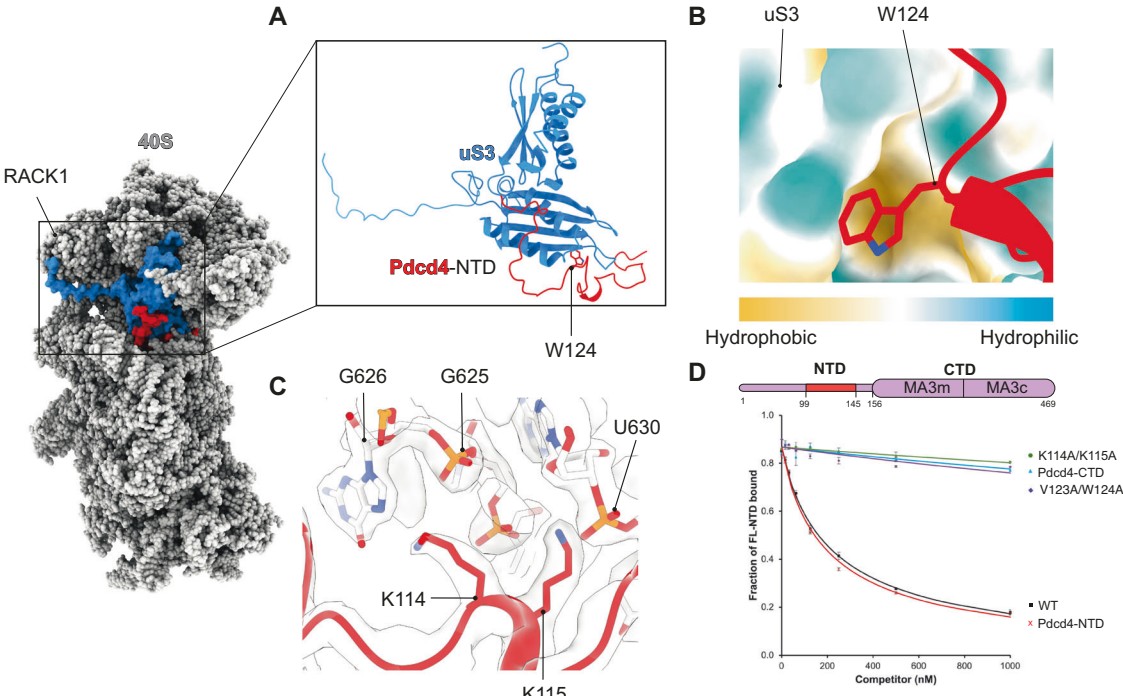

**Fig. 2 | Specific interactions around the entry channel mediate the binding of Pdcd4-NTD to the 40S. A** Pdcd4-NTD binds to the ribosomal protein uS3. **B** W124 in Pdcd4-NTD is anchored to a hydrophobic pocket in uS3, and plays a critical role in this interaction. **C** K114 and K115 in Pdcd4-NTD also makes specific interactions with h18 in 18S rRNA. **D** Fluorescence anisotropy competition assay using labeled Pdcd4-NTD similar to Fig. 1B, showing that residues described in **B** and **C** are critical for the binding of Pdcd4 to the 40S. V123A/W124A and K114A/K115A mutants almost completely lose their affinity for the 40S, which is comparable to deletion of the entire NTD (Pdcd4-CTD, residues 157-469). Error bars represent the mean ± SEM ($n = 3$). For the NTD, only two independent experiments were performed. The experiment was repeated at least three times with similar results.

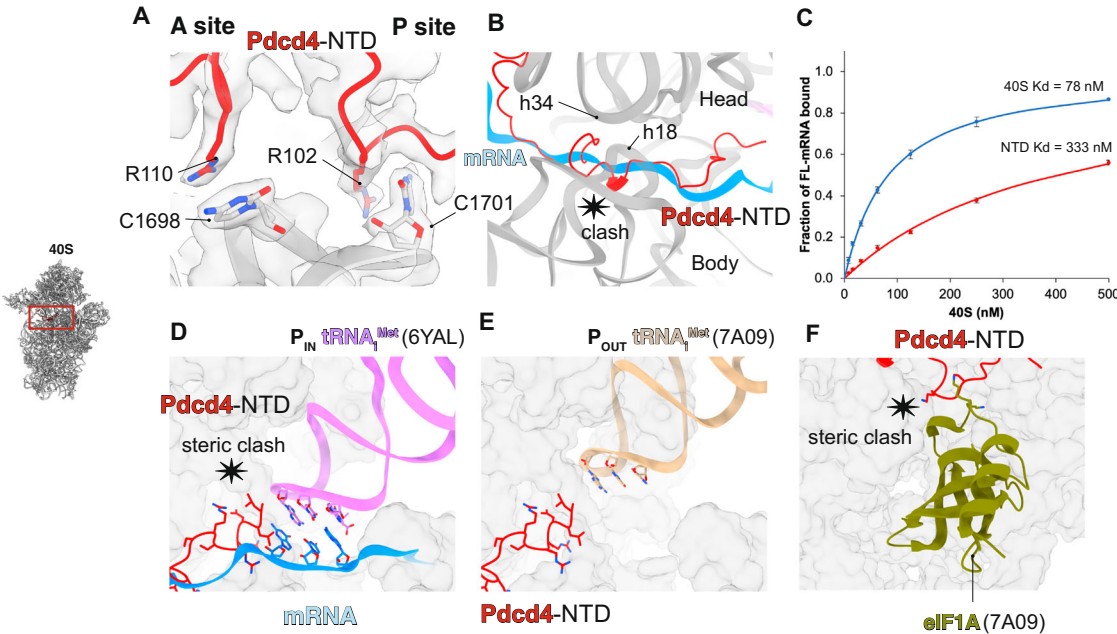

**Fig. 3 | Pdcd4 blocks the decoding center of the 40S subunit. A** Pdcd4-NTD interacts with the universally conserved decoding bases in the A site and the P site. **B** Superposition with the 48S complexes in closed/P_in state (PDB:6YAL)[24] showing a severe clash between mRNA and Pdcd4-NTD from the entry channel to the P-site. **C** Fluorescence anisotropy binding assay using a fluorescence-labeled 71-nt CAA repeat RNA, showing that Pdcd4-NTD reduces mRNA binding to the 40S. Error bars represent the mean ± SEM ($n = 3$). **D** tRNAi^Met in P_in state (PDB:6YAL)[24] clashes with Pdcd4-NTD, while tRNAi^Met in P_out state (PDB:7A09)[48] (**E**) does not. **F** Superposition of Pdcd4-NTD with the structure of either the head or body of the 43S in open/P_out state, showing a possible minor clash with eIF1A only when Pdcd4-NTD is modeled to the body.

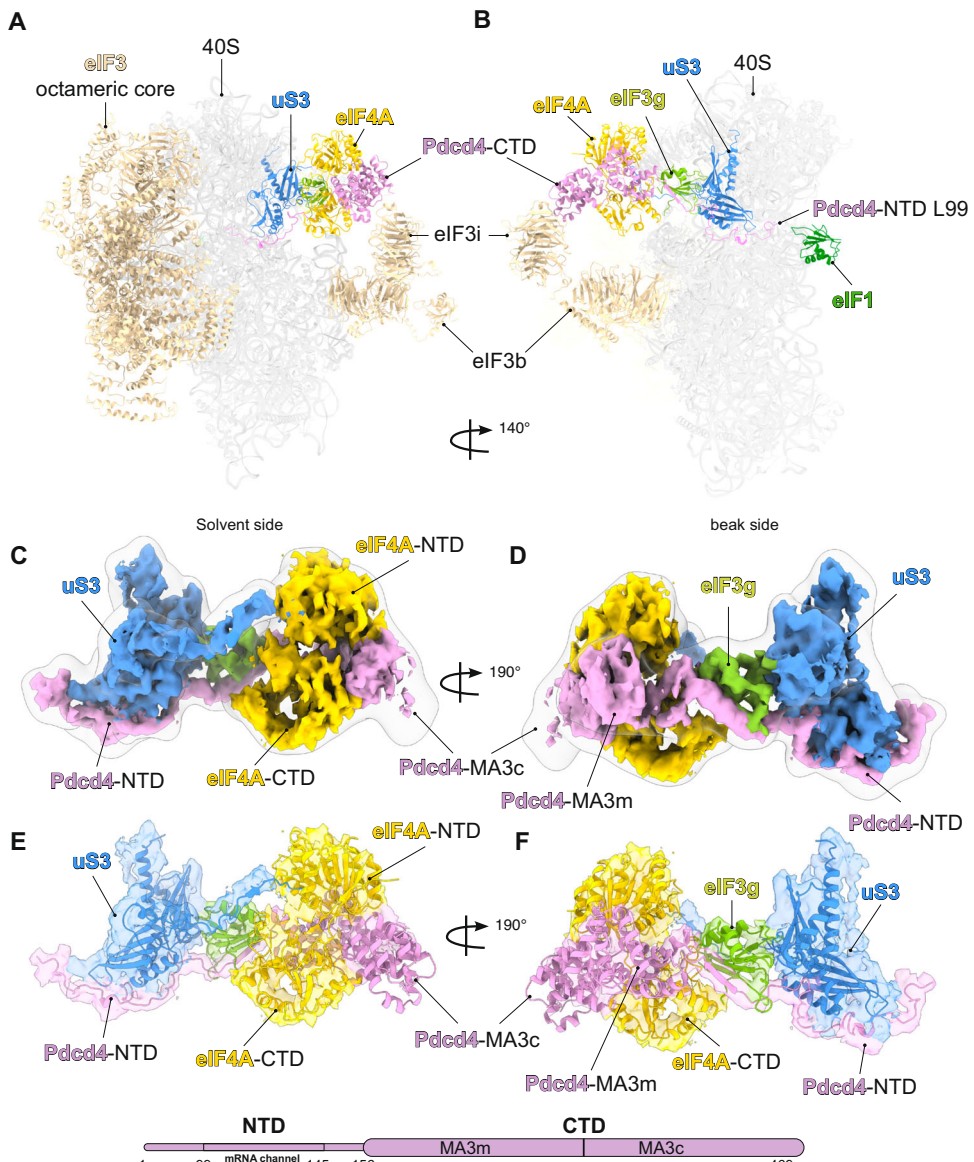

**Fig. 4 | Cryo-EM 3D reconstruction of a human Pdcd4-40S-eIF4A-eIF3-eIF1 complex. A, B** Overall view of the cryo-EM structure shown in different orientations to highlight Pdcd4. **C, D** Cryo-EM map to highlight the density at the mRNA entry site that we assigned to Pdcd4 and eIF4A. The sharped map is docked into a map low-pass filtered to 15 Å (transparent silver). **E, F** Models of Pdcd4-CTD and eIF4A (PDB:2ZU6, where one of eIF4As is omitted)[4] fitted into the cryo-EM map to highlight the interaction of Pdcd4 with eIF4A, uS3, and eIF3g-RRM.

(-75%, Supplementary Fig. 6) or Pdcd4-40S identical to the above structure (16%), cryo-EM analysis reveals two novel classes of particles containing 40S, eIF1, eIF3, and Pdcd4 in the presence (0.9%) or absence (2.7%) of eIF4A (Supplementary Fig. 1). Since the structure in the absence of eIF4A only shows a part of Pdcd4-NTD identical to one found in the Pdcd4-40S structure, we hereafter only describe details of the structure containing eIF4A.

In this structure (overall resolution of 3.9 Å), Pdcd4-NTD binds to the mRNA channel and induces the 40S head swivel in a manner identical to the Pdcd4-40S structure. An extra density occupying a space between eIF3i and eIF3g-RRM is now observed (Supplementary Fig. 7A–C), which directly connects to Pdcd4-NTD in the mRNA entry channel. A local resolution of 7–12 Å (Supplementary Fig. 2) allows us to attribute this density to Pdcd4-CTD binding to a single eIF4A, in a manner corresponding to one of two eIF4A binding forms observed in the crystal structures of a Pdcd4-CTD-eIF4A complex[4,8] (Fig. 4A–F and Supplementary Figs. 7D–G and 8A, B). In this conformation, two MA3

domains in Pdcd4-CTD bind to the mRNA binding surface of eIF4A, thereby stabilizing eIF4A in an inactive open conformation. We recently identified a second molecule of eIF4A[16] bound to the 48S at the mRNA entry site in addition to eIF4A bound at the exit site as a part of eIF4F[25]. We proposed that the mRNA entry channel bound eIF4A likely unwinds structures of an incoming mRNA at the leading edge of the complex to promote mRNA recruitment and scanning. The Pdcd4-CTD positioned at the mRNA entry site in the current structure, therefore, suggests that it targets this eIF4A to inhibit its helicase activity. Upon binding to Pdcd4-CTD, we observe that eIF4A adopts an almost 160° rotation from its position in the 48S structure, now binding to the opposite side of Pdcd4-CTD and making fewer interactions with the other part of the complex (Supplementary Fig. 8C, D).

In addition to Pdcd4-CTD and eIF4A, we observe the C-terminal region of Pdcd4-NTD connecting to the Pdcd4-CTD. Following W124 making the crucial interaction with uS3 at the mRNA entry channel

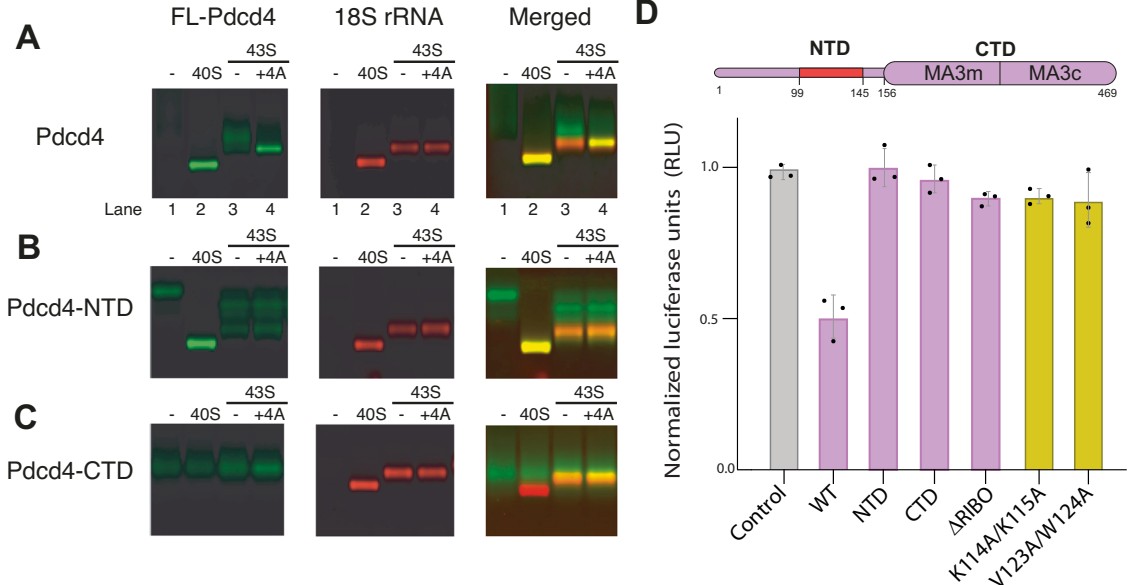

**Fig. 5 | Both N- and C-terminal domains are needed for translational inhibition.**
**A–C** Native gel shift assay with fluorescence labeled-Pdcd4 (green), either using the full-length Pdcd4, Pdcd4-NTD (residues 1-156), or Pdcd4-CTD (residues 157-469), for different 40S ribosomal subunit complexes (stained with ethidium bromide, shown in red). Both FL-Pdcd4 and FL-Pdcd4-NTD tightly co-migrate with the 40S (lane 2), and weakly with 43S (lane 3, <20% binding compared to lane 2). Only FL-Pdcd4 shows tight binding to the 43S in the presence of eIF4A (lane 4, see also Supplementary Fig. 9C). Note that FL-Pdcd4-NTD shows at least three extra non-ribosomal bands in lanes 3–4, which are not observed for FL-Pdcd4-NTD binding to 40S or eIFs independently, leaving their exact nature yet to be determined. FL-Pdcd4-CTD does not co-migrate with any ribosomal complexes, even in the presence of eIF4A. The presence of excess eIF4A causes tightening of the FL-Pdcd4-CTD band without appreciable shift (lane 4), most likely due to the interaction with free eIF4A outside the ribosome. Note the apparent overlap of FL-Pdcd4-CTD band with the 43S does not indicate they interact with each other, as they do not comigrate when the 43S band is shifted by further addition of eIF4F-mRNA (Supplementary Data Fig. 9B). **D** In vitro translation assay in HEK293T cells lysate using an mRNA encoding a firefly luciferase reporter sequence with β-globin 5′ UTR. The relative luciferase activity was measured in the presence of 250 nM Pdcd4 or its variant and normalized to a control performed without Pdcd4. ΔRibo mutant lacks the mRNA binding domain (residues 102–134) in the N-terminal domain. Error bars represent the mean ± SEM ($n = 3$). The experiment was repeated at least three times with similar results.

(Fig. 2B), Pdcd4-NTD binds along uS3 outside the mRNA entry channel toward the eIF3g-RRM (Fig. 4C–F and Supplementary Fig. 8A, B). Notably, the secondary structure prediction and the cryo-EM map reveal that Pdcd4-NTD forms an intermolecular antiparallel β-sheet (Supplementary Fig. 7F–H) with the eIF3g-RRM prior to connecting with the MA3m domain in Pdcd4-CTD, which could be important in stabilizing the MA3m in its current position (Fig. 4E, F). Crystal and NMR structures of the free Pdcd4-CTD suggested that MA3m and MA3c domains are connected with a flexible loop, which undergo a significant change in their relative orientations upon binding to eIF4A[7,8]. Pdcd4-CTD in our structure shows the conformation essentially identical to this eIF4A bound form, suggesting that MA3c is stabilized in the current position by eIF4A. In this position, MA3c appears to make a minor contact with eIF3i, although we note that the density around this region is rather weak (Supplementary Fig. 7A–C). Through this contact, eIF3i could sterically hinder eIF4A binding to MA3c in a manner observed as an alternative binding mode in the previous crystal structures (Supplementary Fig. 8A, B), although we note that this form of eIF4A binding does not seem to occur in solution[7]. Interestingly, as Pdcd4-CTD bridges a gap between eIF3g and eIF3i, a closed ring is formed by eIF3b-eIF3i-Pdcd4-eIF3g between the 40S head and body (Fig. 4A, B), which might interfere with the path of mRNA. The Pdcd4-CTD also partially overlaps with the entry channel binding site of eIF4A in the 48S, and thus could serve to prevent another eIF4A binding to the vacant site while sequestering eIF4A (Supplementary Fig. 8A–D). Overall, outside the entry channel, Pdcd4-NTD extensively interacts with the uS3 and eIF3g on the 40S head, while Pdcd4-CTD makes only a weak contact with eIF3i. This suggests that Pdcd4-CTD is predominantly tethered to the 40S head by Pdcd4-NTD through the intermolecular β-sheet with eIF3g. Therefore, the position of Pdcd4-

CTD seems to be highly dependent on the conformation of the 40S head (Supplementary Fig. 8E, F).

**Translational control by Pdcd4 needs both C- and N-terminal domains**
To further dissect the effect of eIFs on the interaction of Pdcd4 with 40S complexes, we performed native gel assays (NGA) using fluorescence-labeled Pdcd4 (FL-Pdcd4), FL-Pdcd4-NTD, and FL-Pdcd4-CTD. As suggested in Fig. 1A, Pdcd4 and Pdcd4-NTD, but not Pdcd4-CTD, can directly bind to and comigrate with the 40S (Fig. 5A–C, lane 2). Consistent with the affinity of Pdcd4-NTD measured in the anisotropy binding assay (Supplementary Fig. 5), these interactions are significantly weakened in the 43S (Fig. 5A, B, lane 3 and Supplementary Fig. 9A).

In our structure, Pdcd4-CTD is located near eIF3i, and interacts with eIF4A at this position independently of other eIF4F subunits (Fig. 4A, B and Supplementary Fig. 7A–C). We therefore tested if eIF4A can promote Pdcd4 binding to 43S. Strikingly, NGA shows that the addition of eIF4A appreciably stabilizes the binding of Pdcd4 to the 43S even when other eIF4F subunits are absent (Fig. 5A, lane 4). Unexpectedly, despite the strong anticooperativity with eIF1A and tRNAi[Met] for the Pdcd4-NTD (Supplementary Fig. 5), Pdcd4 stably coexists with eIF1A and the eIF2 ternary complex on the 43S in the presence of eIF4A (Supplementary Fig. 9C). In contrast, Pdcd4-NTD does not display this cooperative effect with eIF4A (Fig. 5B, lane 4), suggesting that this additional stabilization is mediated by the interaction between Pdcd4-CTD and eIF4A at the mRNA entry site in the 43S. We observe no shift in the positioning of Pdcd4-CTD on the gel even in the presence of eIF4A (Fig. 5C and Supplementary Fig. 9B), suggesting that stable binding of Pdcd4 to the ribosomal complex is

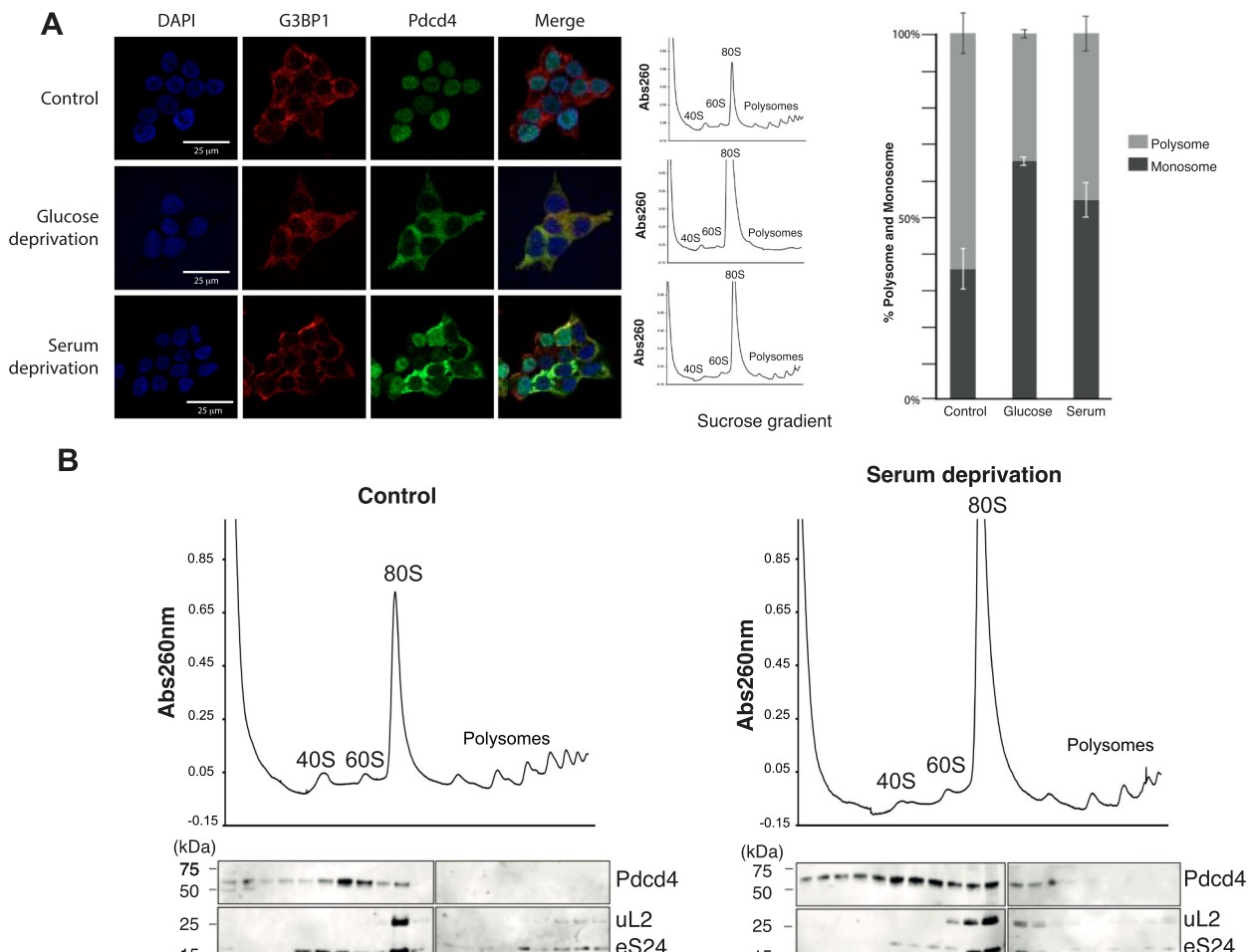

**Fig. 6 | Subcellular localization of Pdcd4 and translational control.**
**A** Immunofluorescence of HEK293T cells shows that the Pdcd4 (green) is mainly colocalized in nucleus (blue) under normal growth, which migrates to cytoplasm under nutrient deprivation, resulting in a lower polysome/monosome ratio. The scale bar is 25 μm. Error bars represent the mean ± SEM ($n = 3$). **B** Sucrose gradient fractionation showing an enrichment of Pdcd4 comigrating with the ribosomal complexes under glucose starvation. The western blot indicated that Pdcd4 co-migrates mainly with the 40S (ribosomal protein eS24) and 80S ribosome (ribosomal proteins eS24 and uL2). The experiment was repeated at least three times with similar results.

highly dependent on both its NTD and CTD. Importantly, Pdcd4 mutations that disrupt the interaction between Pdcd4-NTD and the mRNA entry channel in the 40S complex (K114A/K115A, and V123A/W124A, Fig. 2D) also fail to compete off FL-Pdcd4 from the 43S in the presence of eIF4A (Supplementary Fig. 9D). This suggests that this fundamental interaction remains important for Pdcd4 to stably bind the 43S in the presence of eIF4A.

To further confirm if both Pdcd4-NTD and Pdcd4-CTD are important for translation inhibition, we conducted in vitro translation assays in HEK293T cell lysates using a capped mRNA containing a beta-globin 5′ UTR followed by a firefly luciferase (FLuc) coding sequence. Pdcd4 can inhibit translation at a concentration as low as 250 nM (Supplementary Fig. 10). Indeed, upon the addition of 250 nM purified Pdcd4, translation is inhibited by 50% (Fig. 5D). Consistent with our structure and binding assay, Pdcd4-NTD, Pdcd4-CTD, ΔRibo (a mutant lacking the mRNA channel binding region) and 40S-binding defective mutants (Fig. 2D; Supplementary Fig. 9D) are unable to inhibit translation, at least at 250 nM (Fig. 5D). However, at high concentration (500 nM), all the mutants harboring intact Pdcd4-CTD (Pdcd4-CTD, ΔRibo, 40S-binding defective mutants) can inhibit translation by ~30%, while Pdcd4-NTD lacking entire CTD is unable to inhibit even at 8 μM (Supplementary Fig. 11A, B). Given that these N-terminal domain mutants cannot stably bind to the 40S ribosomal complex, it is possible that the concentration-dependent inhibition observed here does

not occur at the mRNA entry site but may inhibit the pool of eIF4A or eIF4F in the lysate as proposed before[10]. Such a mechanism of inhibition would be consistent with previous studies suggesting that Pdcd4-NTD is not always required for translation inhibition in cells[26,27].

**Nuclear export of Pdcd4 is associated with translational control**
Pdcd4 is predominantly localized in the nucleus under normal condition[11]. However, during cellular stress, it is exported to the cytoplasm[11] where it plays an important role in stress granule formation[28,29]. Whether Pdcd4 is involved in translational control during the stress response is unknown. Previously, the availability of Pdcd4 in the cytoplasm was shown to be regulated through nuclear-cytoplasmic shuttling in tumor cell lines[29,30], where Pdcd4 is downregulated[17] (Supplementary Fig. 12). Here, we used nontumor HEK293T cells and confirmed that Pdcd4 also mainly localizes in the nucleus under normal growth conditions but is exported into the cytoplasm upon serum or glucose deprivation for 24 h (Fig. 6A). Interestingly, we see an increase in the amount of Pdcd4 comigrating with both 40S and the 80S ribosomal complexes, as well as a decrease in total polysomes in sucrose gradients (Fig. 6A, B), which suggests a model whereby under metabolic stress, Pdcd4 shuttles from the nucleus to the cytoplasm, where it binds the ribosome and inhibits translation (Fig. 7).

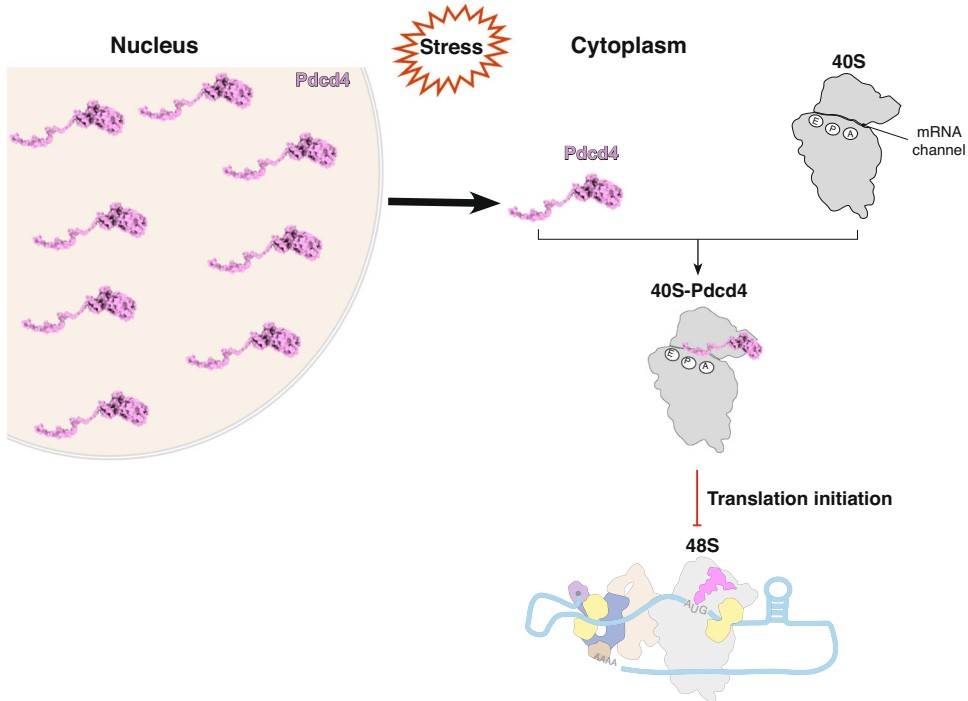

**Fig. 7 | Model for translational control by Pdcd4.** Pdcd4 binds at the mRNA entry site in the 40 S, where it inserts the N-terminal domain into the mRNA channel. Given the high similarity between Pdcd4 and SERBP1, it is likely that Pdcd4 may inhibit translation by blocking the mRNA channel and protecting the ribosome under nutrient deprivation. Additionally, Pdcd4 can inhibit translation by blocking the helicase activity of eIF4A at the leading edge of the ribosome, thereby inhibiting recruitment and scanning, especially during the translation of mRNAs with structured 5′ UTR.

## Discussion

In this study, we determined the structures of human Pdcd4 bound to 40S and 40S-eIF4A-eIF3-eIF1 complex. The position of Pdcd4-CTD in complex with eIF4A at the mRNA entry site agrees well with our recent structure of a human 48S that revealed how a second eIF4A helicase is positioned at the mRNA entry channel in addition to the eIF4A helicase that is part of the eIF4F complex at the exit site[16]. The eIF4A helicase molecule at the entry site would likely be involved in the unwinding of the 5′ UTR of mRNA during recruitment and scanning, as it is located downstream of the mRNA channel. In our model, Pdcd4-CTD is tethered to the 40S head by the Pdcd4-NTD and sequesters eIF4A from its binding site at the mRNA entry site to inhibit its helicase activity (Fig. 7). The inhibition of the entry channel eIF4A provides a mechanism to explain why Pdcd4 preferentially inhibits translation of mRNAs with structured 5′ UTRs[5].

Our NGA data shows that eIF4A stabilizes the binding of full-length Pdcd4, but not Pdcd4-CTD (Fig. 5A–C), suggesting that the interaction between Pdcd4-NTD and the 40S subunit mRNA binding channel plays an important role in stabilizing Pdcd4 on the 43S. Since eIF1A and the eIF2 ternary complex can stably coexist with Pdcd4 on the 43S in the presence of eIF4A (Supplementary Fig. 9C), we anticipate that either the Pdcd4-NTD or 43S (or both) may adopt an alternative conformation to accommodate each other. It is likely that the 43S remains in the open/$P_{out}$ state to accommodate eIF1A and the eIF2 ternary complex, while Pdcd4-NTD might adopt an unknown conformation to be compatible with eIF1A and tRNA$_i^{Met}$ at the A- and P-sites (Fig. 3D–F). Nevertheless, it is important to note that V123A/W124A and K114A/K115A mutants defective in 40S binding are also defective in 43S binding in the presence of eIF4A (Supplementary Fig. 9D). This suggests that these key NTD interactions with the 40S mRNA entry channel observed in the current structure are likely preserved in the interaction with 43S in the presence of eIF4A. This finding is consistent with an earlier study indicating that the same mutants fail to cosediment with the initiation complex[12]. We note, however, that this previous study suggested that these residues promote the binding to the complex by interacting with mRNA or PABP. Our study now demonstrates that these specific residues are indeed critical for the direct binding of Pdcd4 to the 40S mRNA binding channel.

Outside the entry channel, Pdcd4-NTD extensively interacts with uS3 and eIF3g on the 40S head. The formation of an intermolecular β-sheet with the eIF3g-RRM appears to be a key interaction to stabilize the position of MA3m, while eIF4A binding stabilizes the orientation of MA3c. In contrast, Pdcd4-CTD-eIF4A only makes a minor contact with eIF3i, suggesting that Pdcd4-CTD is largely tethered to the 40S head through the interaction mediated by Pdcd4-NTD. Accordingly, when Pdcd4 is modeled onto the uS3 and eIF3g-RRM in the open or closed 43S/48S, it is moved further away from eIF3i (Supplementary Fig. 8E, F). Despite the strong cooperativity between Pdcd4 and eIF4A that we observe in solution, Pdcd4 is completely disordered outside the entry channel in a vast majority of particles (Supplementary Fig. 1). This implies an inherently mobile nature of Pdcd4-CTD on the ribosome. Nevertheless, the current structure agrees well with the cooperativity between Pdcd4 and eIF4A that occurs at the entry site on the initiation complex.

It remains to be determined if Pdcd4 can inhibit the cap-binding complex eIF4F as suggested previously[10]. Consistent with this possibility, Pdcd4-CTD on its own can inhibit translation in cells and in lysate at high concentrations (Supplementary Fig. 11B)[26,27], despite its inability to stably bind to the 43S in the presence of eIF4A (Fig. 5C). However, there is a discrepancy as to whether and how MA3 domains on their own inhibits translation[4]. In our binding assay, we could not observe any evidence to suggest Pdcd4-CTD binding to the 48S nor inhibition of eIF4F binding to the 43S by Pdcd4-CTD (Supplementary Fig. 9B). Future work using precise assays that can distinguish the activities and interactions of eIF4A/eIF4F/Pdcd4 at the entry and exit sites will help to resolve this question.

Besides initiation, high local structural and sequential similarities with ribosome hibernating factors Hapb4/SERBP1/Stm1 (Supplementary Fig. 4B, C) raise a possibility that the specific conformations of the 40S and Pdcd4-NTD in the current structure reflects Pdcd4 acting in a manner similar to these proteins. Consistent with this possibility, we observe a substantial proportion of Pdcd4 comigrating with the 80S ribosomes in sucrose gradient, which is further enhanced upon nutrient deprivation (Fig. 6A, B). Pdcd4 is also suggested to be methylated at R110 under starvation, leading to better cell survivability[30]. As R110 sidechain stacks with a C1698 base at the A-site in our structure (Fig. 3A), its methylation could alter the Pdcd4-NTD interaction at this position by disruption of a planer structure of the R110 side chain, which may result in altered distribution of Pdcd4 in different pathways. Interestingly, SERBP1[20,22] or its yeast homolog Stm1[19] can also bind to ribosomes under metabolic stress, which allows the preservation of the 80S ribosome until nutrient recovery[31]. This process is similar to that of Habp4 in dormant ribosomes in eggs[18]. Alternatively, Pdcd4 binding to the 80S may be relevant to translation inhibition during elongation or promotion of translation termination as suggested previously[17,32].

Taken together, our structure and biochemical data provide critical insights into translational control by the human tumor suppressor protein Pdcd4 and shed light on the role of NTD and CTD in inhibiting translation via direct interactions with the 43S and eIF4A.

During a review of this paper, another study was published[33], which describes a similar structure of Pdcd4-40S-eIF4A-eIF3-eIF1 complex that is consistent with our work.

## Methods

### Purification of human Pdcd4, eIFs, and mRNA
Wild-type Pdcd4 (WT), full-length Pdcd4 with specific point mutations, Pdcd4 lacking N-terminal residues 1 to 134 (Pdcd4-CTD), Pdcd4 lacking residues 101-134 (ΔRibo), and Pdcd4 lacking residues 157 to 469 (Pdcd4-NTD) all tagged with N-terminal His- and MBP-tags along with a TEV protease cleavage site, were expressed in BL21 (DE3) cells at 18 °C overnight. For the native gel assay, we used a Pdcd4-CTD lacking residues 1 to 156. WT and mutant proteins were initially purified using nickel-nitrilotriacetic acid agarose (Ni-NTA) resin (QIAgen). Subsequently, tags were cleaved overnight with TEV protease and removed by passing the samples through a Ni-NTA resin. Following tag removal, the proteins underwent further purification by passing through a Mono S (16/600) column (Pharmacia). We used a gradient of 50-500 mM KCl to elute the proteins. Finally, we performed a polishing step by further purifying the proteins through gel filtration chromatography, utilizing a Superdex 200 HiLoad 16/600 column (Cytiva). For truncated Pdcd4 proteins, we did not perform any further purification steps after tag removal. Ribosome, eIFs and PABP were purified as described previously[34–37]. The mRNA used for cryo-EM was purified as described previously[16].

### In vitro translation
In vitro translation was performed using an in-house HEK293T cell lysate prepared as essentially described before[38]. Briefly, the cells were grown to a count of ~3 million cells/ml at 99% viability. After harvest, cells were washed with ice-cold PBS, and lysed in a hypotonic buffer (10 mM HEPES pH 7.5, 10 mM potassium acetate, 1.5 mM magnesium acetate, and 2 mM DTT). The lysate was then dialyzed for 2 h in a dialysis buffer (20 mM HEPES pH 7.5, 100 mM potassium acetate, 2 mM magnesium acetate, and 1 mM DTT). After dialysis, a total protein concentration was adjusted to 10 mg/ml or kept at the original concentration of 16 mg/ml. The in vitro translation reaction was performed by mixing 4 mg/ml lysate, 0.8 U/µl RNasin Ribonuclease Inhibitor, 0.6 mM DTT, 60 mM potassium acetate, 1.2 mM magnesium acetate, 13 mM HEPES-KOH pH 7.4, 400 nM spermidine, 0.04 mg/mL creatine kinase, 12 mM creatine phosphatase, 0.1 mg/mL pig liver

tRNA, 1 mM ATP, 1 mM GTP, and 40 µM amino acids mixture. The translation reaction was incubated at 30 °C for 5 min, and then mixed with a various amount of Pdcd4 and further incubated at 30 °C for 10 min. To initiate the reaction, 10 ng/ml luciferase reporter mRNA containing β-globin 5′ UTR was added to the mixture, which was then incubated at 30 °C for 1 h. To stop the reaction, 0.22 mM cycloheximide was added, and tubes were placed on ice. To measure the luciferase activity, 9 µl translation reaction was mixed with 45 µl luciferase activity reagent (Promega), and the luminescence was measured.

### Immunofluorescence
HEK293T cells were cultured in DMEM supplemented with 10% FBS (Gibco). After 24 h, media was changed to either fresh DMEM supplemented with 10% FBS, low glucose DMEM (Gibco) supplemented with 5 mM L-glutamine, or serum free media, which was further cultured for 18 h. For immunofluorescence, cells were fixed with 4% paraformaldehyde (PFA) for 15 min at 37 °C. After three washes with PBS, cells were permeabilized with 0.1% Triton X-100 in PBS for 5 min at 4 °C, and blocked with 1 % BSA in PBS for 30 min. Cells were incubated with primary antibodies (anti-G3BP1, BD Bioscience 611126, anti-Pdcd4, Abcam ab51-495, diluted to 1:200) in 0.2 % BSA in PBS for 1 h at room temperature. After washing, cells were incubated with secondary antibodies (Alexa-conjugated IgG H + L series, Thermofisher, diluted to 1:500) and 1 µg/mL DAPI (Sigma) in 0.2 % BSA in PBS for 1 h. Cells were washed with PBS before mounting with ProLong Gold anti-fade mountant (ThermoFisher). Imaging was performed using a Zeiss 780 confocal microscope, while settings were kept constant among experiments.

### Polysome profiling and western blot
Cells grown under serum deprivation were prepared as described above. Cells were washed twice with cold PBS, incubated for 10 min on ice in 600 µL hypotonic lysis buffer (10 mM HEPES pH 7.5, 10 mM potassium acetate, 1.5 mM magnesium acetate, 2 mM DTT and 40 U/µL RNAse OUT) to prevent nuclear lysis, and then lysed by 10 passages through pre-cooled 26G needle. Lysates were clarified by 10 min centrifugation at 12,000×g at 4 °C. Sucrose gradients were prepared with BioComp model 108 Gradient Master using 10% and 50% sucrose solutions in polysomal buffer (30 mM HEPES pH 7.5, 100 mM potassium acetate, 5 mM magnesium acetate, 1 mM DTT). Lysates were loaded into the 10–50% sucrose gradient and centrifuged at 280,000×g for 4 h at 4 °C. Fractions were collected using ISCO Model 160 gradient fractionation system, while 254 nm absorbance was monitored to record the polysome profile. Proteins in each fraction were precipitated by adding 3 volumes of pre-cooled acetone, incubated at −20 °C for 4 h, and centrifuge at 12,000×g for 30 min. The pellets were resuspended in 40 µL 1X LDS sample buffer (Invitrogen), and analyzed by western blotting with antibodies against RPL8 (Abcam - ab169538), RPS24 (Abcam - ab196652), and Pdcd4 (Abcam- ab51-495).

### Native Gel Assay for Pdcd4
The N-terminal domain (NTD, residues 1-156) of Pdcd4 was expressed in BL21 (DE3) cells as a fusion protein with a His$_6$-Maltose-Binding Protein (His$_6$-MBP) tag. It was labeled at an inherent single cysteine, Cys C150, with fluorescein-5-maleimide, following previously established protocols[37]. In brief, the protein was purified using Ni-NTA resin, cleaved with TEV protease, further purified with a Mono Q column, and subsequently used for the labeling reaction. The full-length and C-terminal domain of Pdcd4 (residues 157-469) were expressed in BL21 (DE3) cells as fusion proteins with a C-terminal His$_6$-tagged Mxe GyrA intein. As described in our previous work, these proteins were labeled using an expressed peptide ligation method with a fluorescein-modified Cys-Lys dipeptide[25]. The purification process involved Ni-NTA resin, cleavage with 0.5 M sodium 2-mercaptoethanesulfonate (MESNA), precipitation with 3 M ammonium sulfate, additional Ni-NTA resin purification, Q Sepharose resin purification (GE Healthcare), and

subsequently used for the labeling reaction. Following the labeling, the excess unreacted dye was removed from each protein by passing it through a 600 µl Sephadex G-25 Fine resin (GE Healthcare), pre-equilibrated with a buffer containing 20 mM Hepes-K pH 7.5, 200 mM KCl, 10% glycerol, 1 mM DTT, and 0.1% Tween 20. The labeling efficiencies were typically approximately 60%, 90%, and 40% for the full-length, NTD, and CTD, respectively.

To investigate a direct interaction between the labeled Pdcd4 and a ribosomal complex, we incubated 100 nM of labeled protein at 37 °C for 10 minutes with either 250 nM 40S, 43S (250 nM 40S + 700 nM eIFs 1, 1A, 5, 3j, 300 nM eIF2-TC, and eIF3), or 48S (43S + 300 nM eIF4F, containing stoichiometric amounts of eIFs $4G_{165-1599}$ and 4 A, 300 nM eIF4B, 500 nM eIF4E, and 500 nM 71-nt CAA repeat RNA), in the presence or absence of an additional 500 nM eIF4A. Note that incubation of the final mixture at 37 °C for 10 minutes appears to be sufficient to reach equilibrium. Therefore, an order of mixing, for example, preincubation of FL-Pdcd4 with 40S before adding eIFs or an addition of FL-Pdcd4 to the preformed 43S, does not make any difference in the outcome of NGA. The incubation buffer consisted of 20 mM Tris-acetate pH 7.5, 70 mM KCl, 2 mM $MgCl_2$, 0.1 mM spermidine, 1 mM DTT, 10% glycerol, 0.1 mg/ml BSA, and 0.5 mM ATP-Mg. For competition assays, the reaction also included 3 µM unlabeled Pdcd4, as indicated in each figure. Subsequently, each 10 µl reaction was further incubated on ice for 5 minutes before being loaded onto a 1% agarose gel in THEM Buffer (34 mM Tris, 57 mM HEPES, 0.1 mM EDTA, 2 mM $MgCl_2$). Electrophoresis was carried out for ~90 min at 100 V at 4 °C. Gels were imaged to visualize the fluorescein-labeled Pdcd4 using an Azure Sapphire Biomolecular Imager (Azure Biosystems), and then stained with ethidium bromide to directly observe the 40S subunit.

### Fluorescence anisotropy binding assay

The fluorescence anisotropy binding and competition assays were conducted following established procedures[37]. 71-nt CAA repeat RNA labeled at the 3′-end (CAA71-3FL) was also prepared as described therein. FL-Pdcd4-NTD was used instead of the FL-Pdcd4 for anisotropy assays, because FL-Pdcd4 does not give a sufficient anisotropy signal upon 40S binding. In binding assay, a solution containing 20 nM of FL-Pdcd4-NTD or CAA71-3FL was mixed with the various amount of 40S subunit as indicated, in the presence or absence of 43S components (1 µM eIFs 1, 1A, 5, 550 nM eIF2-TC and eIF3) and 1 µM Pdcd4-NTD in the same buffer used for the native gel assay. For the competition assay, 20 nM FL-Pdcd4-NTD + 50 nM 40S was competed off by various amounts of unlabeled Pdcd4 or its variants as indicated. The reaction, consisting of 20 µl of the mixture, was initially incubated at 37 °C for 10 min and then incubated for over 20 minutes at room temperature before measurements were taken. Similar to NGA above, this incubation appears to be sufficient to reach equilibrium, where an order of mixing does not appreciably affect the result. Measured anisotropy was converted to fraction bound, and fitted with a quadratic equation curve to calculate a dissociation constant.

### In vitro reconstitution of Pdcd4-40S ribosomal complexes for cryo-EM

We reconstituted the Pdcd4-40S complex by mixing 0.5 µM 40S with 1 µM Pdcd4, eIF1, and eIF1A to a final volume of 25 µl in a buffer containing 97 mM potassium acetate, 2.5 mM magnesium acetate, 3% glycerol, 0.1 mM spermidine, 1 mM DTT. The 48S was reconstituted as described before[16]. However, here the 43S was incubated with 1 µM Pdcd4 at 30 °C for 10 min before mixing with 1 µM eIF4F-mRNA, 1 µM eIF4B, 1 µM eIF4A, and 1 µM PABP in a 25 µl reaction. The reaction mix was incubated at 30 °C for 10 min.

### Cryo-EM grid preparation

To avoid the dissociation of eIFs during grid preparation, the complexes were crosslinked with 1.5 mM BS3 on ice for 45 min. We applied 3 µl of 140 nM 40S ribosomal complexes onto UltrAuFoil R1.2/1.3 300 mesh gold grids pre-covered with graphene oxide (Sigma) suspension made in-house as described before[16]. The grids were blotted using filter paper for 7 seconds at 4 °C under 100% humidity using Vitro Mark IV (Thermo Fisher Scientific). Subsequently, they were promptly plunged into liquid ethane at 93 K using a precision cryostat system developed at the MRC LMB[39].

### Cryo-EM data collection and image processing

Data collection was carried out using Titan Krios microscopes (ThermoFisher) equipped with a K3 direct electron detector camera (Gatan) at a magnification of 105,000x and at pixel sizes of 0.86 Å-pix$^{-1}$ using EPU software, super-resolution counting mode using a Bio-quantum energy filter (Gatan) (binning 2), faster acquisition mode. We used a defocus ranging from −1.2 µm to −3.0 µm.

The data were processed using RELION 4[40]. To correct for motion, dose-weighted, and gain-corrected, we used the implementation in RELION 4. To estimate the CTF, we used CTFFIND4.1[41]. After 2D classification, we performed reference-based 3D classification after low-pass filtering the reference map to 60 Å. After 3D refinement, Bayesian polishing and CTF Refinement[40], we performed further focus classification to sort particles containing Pdcd4. First, we conducted mask classification on eIF3, followed by mask classification on TC and Pdcd4. A comprehensive cryo-EM data processing workflow is illustrated in Supplementary Fig 1. The final refinement was performed in Relion 5, applying blush regularization[42] for denoising, which allowed us to improve the quality of the map for the Pdcd4-40S-eIF4A-eIF3-eIF1 complex.

### Model building, fitting, and refinement

For model building and refinement, we used the known structure of the human 40S ribosomal complex[25] as a reference for fitting and model building in Coot[43]. The N-terminal domain of Pdcd4 was built de novo using Coot. For the C-terminal region of Pdcd4-NTD, we initially predicted its secondary structure using PSIPRED[44], enabling us to build the beta strand. Given the low local resolution, we modeled this region at the poly-alanine level due to the inability to discern individual side chains. For Pdcd4-eIF4A, we used a known crystal structure[4] for rigid body fitting into the cryo-EM map. The initial fitting was performed using Chimera[45], followed by manual adjustments and refinement in Coot[43]. We used Phenix for real space refinement[46].

### Figures

All figures were made using ChimeraX[47]. Figure 7 was generated using Adobe Illustrator and InDesign. The surface representation of Pdcd4 was produced using ChimeraX.

### Reporting summary

Further information on research design is available in the Nature Portfolio Reporting Summary linked to this article.

## Data availability

Atomic models have been deposited in the Protein Data Bank (PDB) with IDs 9BKD (40S-Pdcd4) and 9BLN (Pdcd4-40S-eIF4A-eIF3-eIF1). EM maps have been uploaded to the Electron Microscopy Data Bank with accession codes EMD-44641 (40S-Pdcd4) [https://www.ebi.ac.uk/emdb/EMD-44641] and EMD-44671 (Pdcd4-40S-eIF4A- eIF3-eIF1) [https://www.ebi.ac.uk/emdb/EMD-44671]. Source data are provided with this paper.

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

## Acknowledgements
We thank the LMB cryo-EM facilities for their support during data collection; J. Grimmett and T. Darling for computing. Funding: J.B.Q. was supported by a FEBS long-term fellowship, the Rogel Cancer Center – (NIH Award Number P30 CA046592), Chan Zuckerberg Initiative, and LSI computer cluster NIH S10OD030275; I.D.L. was supported by an EMBO Postdoctoral Fellowship; V.R. was supported by the UK Medical Research Council (MC_U105184332), a Wellcome Trust Investigator award (WT096570), and the Louis-Jeantet Foundation; C.S.F. was supported by the NIH (grant R01 GM092927 and R35 GM152137) and a seed grant for international collaboration from Global Affairs and the College of Biological Sciences at the University of California, Davis.

## Author contributions
M.S., C.S.F., and Y.G. purified eIFs; J.B.Q. purified the mRNA, assembled the complexes, prepared cryo-EM grids, and performed the cryo-EM data collection and processing, and built atomic models; J.B.Q, P.Z., and Y.D refined and validated the atomic models; J.B.Q and I.D.L performed the in vitro translation assay; I.D.L and L.A. performed the immunofluorescence; I.D.L performed the western blot and polysome profiling. M.S performed the binding and gel shift assays; J.B.Q., I.D.L., M.S., C.S.F., and V.R. wrote the manuscript with input from all authors. V.R. and C.S.F. supervised the project.

## Competing interests
The authors declare no competing interests.
