## [Peer Review File · Nature Communications]

Human tumor suppressor protein Pdcd4 binds at the mRNA entry channel in 40S small ribosomal subunitREVIEWER COMMENTS

Reviewer #1 (Remarks to the Author):

In the present manuscript, Querido et al. elucidate the role of the tumor suppressor protein Pdc4 in translational control in humans by combining cryo-EM structures with functional in vitro and in vivo data. Their 2.6 Å structure of a 40S-Pdc4 complex reveals in detail the role of Pdc4's N-terminal RNA recognition motifs (NTD) in binding and sequestering the mRNA channel and decoding center of the ribosomal 40S subunit to preclude mRNA recruitment, which the authors further support by in vitro assays testing 40S binding. The structure of a 40S-eIF1-eIF3-Pdc4 complex shows a similar arrangement of the Pdc4-NTD and additional density at the mRNA entry that the authors suggest to be Pdc4's two C-terminal MA3 domains (CTD). Based on in silico modeling with the help of previous structures the authors furthermore propose the MA3 domains might interact with an eIF4F-independent eIF4A at the mRNA entry; the two MA3 domains are known to block eIF4A's helicase activity. Native gel shift experiments and in vitro translation assays reveal that both NTD and CTD of Pdc4 are required for 43S binding and translation inhibition, respectively, with eIF4A strongly stabilizing 43S binding. Finally, the authors show that Pdc4 is largely localized in the cellular nucleus, but, upon nutrient deprivation, re-localizes to the cytoplasm, where it binds to 40S and 80S ribosome complexes.

Overall, the manuscript provides important novel insights into the cooperative function of Pdc4's NTD and CTD in inhibiting translation, suggesting a model that Pdc4 binds to 40S dependent on both NTD and CTD in a cooperative manner, and controls translation by blocking either the mRNA channel and/or the helicase activity of an eIF4F-independent eIF4A at the mRNA entry. The work is well described and generally supported by the data. However, there are several points that should be addressed by the authors to warrant publication.

Major points

- 1.a) Cryo-EM image processing and atomic modeling should be explained in more detail, giving specific information.
- b) Computational classification of both cryo-EM data sets should be described in detail in both the methods and in a sorting scheme, showing classification criteria and approaches, masks used for focused classifications, the number of particles going into the distinct classes at each step, and what states the classes represent. This is essential to judge the relevance of the final structures.
- c) In this line, what reasons (maybe also biochemical ones) are there for the low number of particle images (~3%) going into the final 40S-eIF1-eIF3-Pdc4 structure?
- d) For validation of the cryo-EM maps and atomic models, map-model FSCs should be provided and the local resolution map for the weak extra densities at the mRNA entry of the 40S-eIF1-eIF3-Pdc4 structure should be shown.

2. In Figure 4, several regions of the 40S-eIF1-eIF3-Pdcd4 cryo-EM map appear to be rendered with substantially different density thresholds, including eIF1, parts of the eIF3 core, the eIF3b/i module and the potential Pdcd4-CTD at the mRNA entry, which show very scattered densities. The authors should denote how the different segmented densities were rendered in the figure legend, stating specific thresholds and the potential use of narrow masks. For a more realistic depiction, the authors should provide an additional figure showing the 40S-eIF1-eIF3-Pdcd4 structure with the less-well defined regions low-pass filtered to the respective local resolutions. The docking of Pdcd4-CTD should be shown for the low-pass filtered version in the main figure (see also point 3b). The authors should also state the used density thresholds in the legend of Extended Data Fig. 5.

3. The authors suggest the density at the mRNA entry in the 40S-eIF1-eIF3-Pdcd4 structure represents the Pdcd4's MA3 domains. However, the weak density, and the presence of other factors binding in this area such as eIF4A, eIF4B, eIF3g (which is only partially modelled) makes a density-based assignment to a specific protein complex a challenge.

a) It is unclear, if the authors tried to improve density for this crucial area by additional sorting on the mRNA entry area, and the weakly defined eIF3b/i (which may stabilize the Pdcd4-CTD). If yes, this should be clarified in the methods (see point 1b). If not, the authors might try focused classification in combination with new methods such as blush regularization or 3D flexible refinement.

b) Currently, the weak density, as shown in Figure 4, appears to be interpreted at too high resolution, dominated by high-noise levels (see also point 2). For reliable atomic model docking, the density should be low-pass filtered to the local resolution, which appears to be rather at the 12-15 Å level.

c) The main argument for Pdcd4-CTD binding close to the mRNA entry is the direct contact with the well-resolved Pdcd4-NTD (up to residue 145 at the mRNA entry); in absence of additional data it remains unclear, if the density provides the correct location of Pdcd4-CTD. Therefore, the authors should tone down their statements accordingly, e.g. in lines 172-173 "two MA3 domains" should be replaced with "additional density", and in line 309 "as observed in our structure" with "as suggested by our structure".

4. In the methods it should be clarified, whether 43S/48S complexes for native gel assays were assembled in presence of Pdcd4, or if Pdcd4 was added after 43S/48S assembly. In this line, it would be interesting to know (this is no request!), if the authors tried pre-incubation of Pdcd4, or Pdcd4-NTD with vacant 40S subunit before adding eIFs and tRNA to increase binding to the 43S complex, independent of eIF4A?

5. a) In Figure 5A/B co-migration of Pdcd4 with 43S complexes w/o eIF4A, and of Pdcd4-NTD with 43S complexes in general, is hard to recognize. Can the authors exclude, particularly for the full-length Pdcd4, that this results from a smearing of the other additional bands observed under these conditions? The corresponding legend should acknowledge the differences at least by adding a quantitative statement.

b) In Figure 5C, the overlap of Pdcd4-CTD band with 43S prevents interpretation of 43S binding, which is clearer in Extended Data Fig. 9A. Therefore, a reference to the latter in the Figure 5 legend

would be helpful. Moreover, in line 509 using “free eIF4A” instead of “eIF4A on its own” might improve legibility.

6. In Figure 3B, the authors show that Pdc4-NTD might interfere with Pin initiator tRNA binding in the closed 48S initiation complex by a slight steric clash. However, as Pdc4-NTD strongly precludes mRNA binding, and thus 48S formation, the more relevant question is in how far the NTD might interfere with Pout tRNA in human 43S complex, which has no mRNA bound (PDB 7A09). Figure 3B should be replaced accordingly.

7. From Extended Data Figure 4 it is not clear in how far Pdc4-NTD and SERBP1 sequences correlate with their structures on the 40S; this information should be added in the figure.

Minor points

8. In line 133, to be more specific, the authors should replace “the affinity of an RNA”, with “the affinity of an mRNA”.

9. In lines 281-285, the authors might mention their observation of Pdc4 comigrating with 80S ribosomes.

10. In Figure 7 it would be helpful to show the Pdc4 schematic always with the same scaling. Moreover, the figure might be supplemented by information on the nutrient-dependence of Pdc4 localization.

11. It would be helpful for the reader to mention in the sorting scheme for the 40S-eIF1-eIF3-Pdc4 structure (see point 1b) that complexes were obtained in the context of 48S assembly.

12. In several figure legends references to panels are missing, e.g. for Figure 2A, Figure 4C, Extended Data Figure 3B, Extended Data Fig. 4B.

Reviewer #2 (Remarks to the Author):

The tumor suppressor Pdc4 is a translation regulator that consists of two N-terminal RRM motifs and the middle and C-terminal MA3 domains. It was shown that the N-terminal RRM motifs are required for association of Pdc4 with 40S ribosomal complexes, whereas the MA3 domains bind to eIF4A inhibiting its helicase activity. However, the exact mechanism by which Pdc4 influences translation has remained unresolved. Here, Querido et al. report the cryo-EM structures of 40S/Pdc4 and 40S/eIF3/eIF1/Pdc4 complexes to gain more insights into this process. The first structure revealed the ribosomal position of a small N-terminal fragment (a.a. 99-145) of Pdc4 bound in the mRNA channel of the 40S subunit. The second structure additionally contained density positioned at the mRNA entrance (interacting with the eIF3bgi module) which was tentatively attributed to the MA3 domains of Pdc4, even though the resolution was too low to allow modelling of its structure.

Interestingly, the binding site and the structure of the 40S-binding domain of Pdc4 resembled those of ribosomal hibernation factors Serbp1/Stm1p. The functional importance of ribosome-interacting residues in Pdc4 was shown using appropriate mutants in 40S/Pdc4 binding experiments. Importantly, the region of Pdc4 bound in the mRNA channel would clash with eIF1A and the eIF2/GTP-Met-tRNA^{iMet} ternary complex. However, although the Pdc4 N-terminal domain (a.a. 1-156) and full-length protein had comparable affinities for the 40S subunit, it did not inhibit translation, suggesting that the MA3 domains that are known to interact with eIF4A are essential for inhibition. On the other hand, the individual C-terminal MA3 domains alone were also not able to induce potent translation inhibition, indicating that the presence of both N- and C-terminal regions of Pdc4 is required for the Pdc4 function.

The position of the additional density at the mRNA entrance in the 40S/eIF3/eIF1/Pdc4 complex that was attributed to the MA3 domains was compatible with their interaction with eIF4A bound at its second ribosome binding site, which is also at the mRNA entrance. This prompted the authors to investigate the influence of eIF4A on association of Pdc4 with 40S ribosomal complexes. Indeed, using native gel shift assay, the authors were able to demonstrate eIF4A-induced stimulation of binding of full-length Pdc4 to, what the authors claim, are 43S preinitiation complexes (see Major critique).

In conclusion, this is an interesting study, in which the conclusions derived from the rather incomplete cryo-EM structures are supported by biochemical data.

MAJOR CRITIQUE

Regarding the facts that (i) binding of Pdc4 in the mRNA-binding channel would cause a clash with eIF1A and eIF2-ternary complex and (ii) that the authors consistently did not find the class in which Pdc4 is bound to the 40S subunit in the presence of eIF2/Met-tRNA^{iMet} (even though the mixture for 48S complex formation, which yielded the 40S/eIF3/eIF1/Pdc4 complex, by definition did contain eIF2/Met-tRNA^{iMet}) raise the question of the exact composition of 40S complexes to which Pdc4 is bound in Figure 5. These could be not the bona fide 43S complexes, but instead 40S/eIF3/eIF1 complexes, exactly those, which the authors observed in cryo-EM. The authors must repeat these experiments using side by side 43S and 40S/eIF3/eIF1 complexes with/without eIF4A.

It is also surprising that the authors do not detect eIF4A associated with the 40S/eIF3/eIF1/Pdc4 complex, even though ribosomal complexes were stabilized by cross-linking before being applied to the grids. What was the difference in the conditions for complex preparation etc. compared to those used by the same group when they detected eIF4A at the mRNA entrance? In fact, the presumable Pdc4/eIF4A interaction at that location would mutually stabilize ribosomal binding of both proteins.

MINOR COMMENTS

1. It is an exaggeration to claim (line 257) that the structure of human Pdc4 has been determined on the ribosome. The authors were able to model the structure of a fragment corresponding to 46 out of a total of 465 a.a. residues.

2. The authors should explain what an MA3 domain is (line 52).

It would be helpful if Figure 1A showed the borders of the two RRM domains.

3. The authors should comment on their finding that the Pdc4-CTD does not inhibit translation in light of reports that even the C-terminal MA3 domain alone inhibits translation in transfected cells (e.g. Laronde-Leblanc et al, MCB 2007 (PMID: 17060447)) .

4. The title page shows that the institutional affiliation of M. Sokabe and CS Fraser is with the University of Michigan; although UC Davis is listed, none of the authors is affiliated with it.

5. The Methods section has been carelessly prepared, and contains numerous typographical errors, grammatical errors and omissions.

Line 464. The method may be “widely described”, but the authors should nevertheless cite an appropriate reference

Line 465, 502. The quality of English throughout Methods should be improved e.g. “after harvested and washed” (line 465) and “were measure in the nanodrop’ (line 502).

Line 473. What is the “translation reaction buffer”?

Line 475. Correct “b-globin 5c UTR”

Line 484. What is “PFA”?

Lines 486-7, 510. The sources of antibodies should be mentioned.

Line 498. ‘trough’ should be ‘through’

Line 516. Missing reference.

Lines 521-522. Missing reference.

Line 547. The PMID number is not necessary here.

Lines 563, 567. It seems unlikely that the reaction volume was 25ml.

Line 581. The cited reference does not refer to Relion4.

Reviewer #3 (Remarks to the Author):

Pdcd4 inhibits mRNA translation in response to stress, although the molecular mechanism is not entirely understood. This manuscript takes a structural approach to understanding how Pdcd4 binds to and inhibits the ribosome. The authors show that the N-terminal domain of Pdcd4 inserts into the mRNA entry site of the 40S subunit. This likely prevents mRNA binding, given that it would sterically clash with Pdcd4. In addition, this anchors Pdcd4 to the 40S, allowing the C-terminal domain of Pdcd4 to interact with eIF4A and to act.

Overall, I think this manuscript provides useful and interesting insights into the molecular mechanism how Pdcd4 can function. It will be of interest to a wide audience of people studying mRNA translation and stress responses. I found the data solid and convincing. I have only very minor comments:

Minor:

1. Fig 4B-C: it's not clear which is the cryoEM density (grey?) and which is the rigid-body fitting of the known crystal structure of Pdcd4-CTD (pink?).

It would be good to label them.

2. Fig 5, the labeling of lane 1 ("WT") is confusing because for all the other lanes the label describes what is present in addition to Pdcd4, so it should actually say "-". Otherwise it seems like WT ribosomes are added? (Same for 5B-C).

Dear reviewers,

We thank you all very much for all the valuable comments. We have addressed and incorporated all the suggested comments from reviewers into our revised manuscript and our detailed responses to the comments are included below. Since our initial submission, we significantly improved the local resolution in the cryo-EM analysis using a new version of Relion software. This allowed us to identify a new density on our grids corresponding to Pdc4-CTD in complex with eIF4A at the mRNA entry site. Thus, we now focus our description on this new Pdc4-40S-eIF4A-eIF3-eIF1 structure. Accordingly, we have a new section subtitled “Pdc4-CTD interacts with eIF4A at the mRNA entry site” describing the new structure in the results section, in place of the previous section subtitled “A possible location for Pdc4 MA3 domains”. We also significantly revised the Discussion based on the new result. In the light of the new structure and the additional biochemical assays that we carried out, we believe that we have addressed all the reviewer’s comments and have greatly strengthened our manuscript.

We wish to acknowledge that a manuscript was recently published (April 19th) that has significant overlap with our manuscript (PMID: 38641729). In contrast to our reconstituted purified system, their study purified a Pdc4-40S-eIF4A-eIF3-eIF1 complex from cell extracts. These different approaches each have benefits and drawbacks, but we note that the structures obtained are very similar and draw similar conclusions about the mechanism by which Pdc4 inhibits initiation. In contrast to the other manuscript, we highlight that we carried out translation assays and quantitative binding assays using wild type and mutant Pdc4 to rigorously characterize the interaction between Pdc4 and different initiation complexes. We therefore feel strongly that these two manuscripts complement each other and each possess different strengths.

Link to map and models: <https://cloud.mrc-lmb.cam.ac.uk/s/5dytq5bkNbRtRNc>

REVIEWER COMMENTS

Reviewer #1 (Remarks to the Author):

In the present manuscript, Querido et al. elucidate the role of the tumor suppressor protein Pdc4 in translational control in humans by combining cryo-EM structures with functional in vitro and in vivo data. Their 2.6 Å structure of a 40S-Pdc4 complex reveals in detail the role of Pdc4’s N-terminal RNA recognition motifs (NTD) in binding and sequestering the mRNA channel and decoding center of the ribosomal 40S subunit to preclude mRNA recruitment, which the authors further support by in vitro assays testing 40S binding. The structure of a 40S-eIF1-eIF3-Pdc4 complex shows a similar arrangement of the Pdc4-NTD and additional density at the mRNA entry that the authors suggest to be Pdc4’s two C-terminal MA3 domains (CTD). Based on in silico modeling with the help of previous structures the authors furthermore propose the MA3 domains might interact with an eIF4F-independent eIF4A at the mRNA entry; the two MA3 domains are known to block eIF4A’s helicase activity. Native gel shift experiments and in vitro translation assays reveal that both NTD and CTD of Pdc4 are required for 43S binding and translation inhibition, respectively, with eIF4A strongly stabilizing 43S binding. Finally, the

authors show that Pdc4 is largely localized in the cellular nucleus, but, upon nutrient deprivation, re-localizes to the cytoplasm, where it binds to 40S and 80S ribosome complexes. Overall, the manuscript provides important novel insights into the cooperative function of Pdc4's NTD and CTD in inhibiting translation, suggesting a model that Pdc4 binds to 40S dependent on both NTD and CTD in a cooperative manner, and controls translation by blocking either the mRNA channel and/or the helicase activity of an eIF4F-independent eIF4A at the mRNA entry. The work is well described and generally supported by the data. However, there are several points that should be addressed by the authors to warrant publication.

Major points

1.a) Cryo-EM image processing and atomic modeling should be explained in more detail, giving specific information.

In the revised manuscript, we presented a comprehensive cryo-EM data processing workflow in a new Extended Fig. 2, delineating each step in more detail. We also explained the approaches utilized in the materials and methods and elaborated on the atomic model building and refinement.

b) Computational classification of both cryo-EM data sets should be described in detail in both the methods and in a sorting scheme, showing classification criteria and approaches, masks used for focused classifications, the number of particles going into the distinct classes at each step, and what states the classes represent. This is essential to judge the relevance of the final structures.

As mentioned above, we have now extended materials and methods and present a comprehensive cryo-EM data processing workflow.

c) In this line, what reasons (maybe also biochemical ones) are there for the low number of particle images (~3%) going into the final 40S-eIF1-eIF3-Pdc4 structure?

While there can be many different reasons as to why one preferentially obtains specific complexes on grids, we can only make some suggestions as to why we obtain only specific structures in this study.

We have used our NGA assay in the original manuscript and the revised manuscript to determine the relative stability of different initiation complexes that include Pdc4. As shown by NGA, the relative binding of Pdc4 for the 43S-eIF4A is reasonably strong compared to Pdc4 binding to the 40S on its own (Fig. 5A, lanes 4 vs. lane 2). We confirm in the new Extended Fig. 9C that Pdc4 can also bind to the bona fide 43S-eIF4A containing eIF1A and the eIF2 ternary complex, as per reviewer 2's suggestion (detailed below). Interestingly, we have independently confirmed that the addition of eIF4F-mRNA to the 43S does not appreciably affect Pdc4 binding, as determined by NGA (data not shown). Thus, these observations suggest that Pdc4 can stably coexist with eIF1A and tRNA on 43S-eIF4A (or 48S-eIF4A), at least under the condition we use to reconstitute the cryo-EM sample in solution prior to crosslinking and applying to a grid.

While complexes containing Pdc4 appear to be relatively stable on our NGA, the reason for why we only obtain preferential structures is not entirely clear. We note that most complexes observed on the grid appear to be 43S/48S complexes in the absence of Pdc4. Although it is unclear why we could not visualize Pdc4 on the 43S/48S complex, it is entirely possible that Pdc4 has dissociated or that the

conformation of Pdc4 on the 43S becomes less well ordered (see our new Discussion), and thus was less favored by a crosslinker or averaged out during 3D reconstruction.

The Pdc4-40S-eIF1-eIF3 and the new Pdc4-40S-eIF4A-eIF1-eIF3 complexes are minor subpopulations on the grid (~3% and ~1% of the particles) despite them being of similar apparent stability to Pdc4-40S in our NGA. A low percentage is possible given the propensity of Pdc4 to bind to the 40S subunit and induce conformational changes that may alter the binding of other initiation factors, especially eIF1A and eIF2 ternary complex. We note that we have not directly measured binding kinetics of Pdc4 to each complex, or for eIFs to the 40S in the presence of Pdc4, raising the possibility that association and dissociation rates could be significantly altered and therefore limit the complexes obtained.

It is noteworthy to mention that complexes consistent with our structures are present in vivo, as evidenced by a recent study of native Pdc4-ribosomal complexes, published during this revision (PMID: 38641729). In that study, even when using affinity purification, a mere 1.9% of the particles corresponded to the Pdc4-40S-eIF4-eIF3-eIF1 complex.

We hope in future to carry out a kinetic analysis of Pdc4 binding to better characterize how it interacts with different initiation complexes, which may help better understand our observations on grids. Additional details regarding the new NGA data is presented in response to reviewer 2 questions below.

d) For validation of the cryo-EM maps and atomic models, map-model FSCs should be provided and the local resolution map for the weak extra densities at the mRNA entry of the 40S-eIF1-eIF3-Pdc4 structure should be shown.

In the revised manuscript, we incorporated map-model FSC curves and included a figure delineating the local resolution of each relevant region in the complex (Extended Data Fig. 2).

2. In Figure 4, several regions of the 40S-eIF1-eIF3-Pdc4 cryo-EM map appear to be rendered with substantially different density thresholds, including eIF1, parts of the eIF3 core, the eIF3b/i module and the potential Pdc4-CTD at the mRNA entry, which show very scattered densities. The authors should denote how the different segmented densities were rendered in the figure legend, stating specific thresholds and the potential use of narrow masks. For a more realistic depiction, the authors should provide an additional figure showing the 40S-eIF1-eIF3-Pdc4 structure with the less-well defined regions low-pass filtered to the respective local resolutions. The docking of Pdc4-CTD should be shown for the low-pass filtered version in the main figure (see also point 3b). The authors should also state the used density thresholds in the legend of Extended Data Fig. 5.

We have now included details regarding the thresholds used in each figure to enhance clarity within the manuscript. Furthermore, we have fitted the sharpened map into a map filtered at local resolution using or low-pass filtered to 15 Å. Alongside the model-to-map fitting utilizing a sharp map, we introduce a novel figure showing the atomic model fitted within a low-pass filtered map.

3. The authors suggest the density at the mRNA entry in the 40S-eIF1-eIF3-Pdc4 structure represents the Pdc4's MA3 domains. However, the weak density, and the presence of other factors binding in this area such as eIF4A, eIF4B, eIF3g (which is only partially modelled) makes a density-based assignment to a specific protein complex a challenge.

Since our previous manuscript submission, we have taken advantage of a new version of Relion (Relion 5) to enhance data processing. This allowed us to improve the local resolution and identify a density corresponding to Pdc4-CTD in complex with eIF4A at the mRNA entry site. The Pdc4-40S-eIF3-eIF1 complex is likely to be a state where Pdc4-CTD is highly flexible, possibly due to a lack of eIF4A binding. Thus, we have now focused the model building and the discussion on the most relevant complex, Pdc4-40S-eIF4A-eIF3-eIF1. We feel that this new structure strengthens our manuscript and addresses this question.

a) It is unclear, if the authors tried to improve density for this crucial area by additional sorting on the mRNA entry area, and the weakly defined eIF3b/i (which may stabilize the Pdc4-CTD). If yes, this should be clarified in the methods (see point 1b). If not, the authors might try focused classification in combination with new methods such as blush regularization or 3D flexible refinement.

We have now used blush regularization in Relion 5, which likely improved the local resolution. The revised Figure S1 elucidates detailed descriptions of the classification and particle sorting procedures.

b) Currently, the weak density, as shown in Figure 4, appears to be interpreted at too high resolution, dominated by high-noise levels (see also point 2). For reliable atomic model docking, the density should be low-pass filtered to the local resolution, which appears to be rather at the 12-15 Å level.

We have now provided a model for map fitting using a map filtered at local resolution or lowpass filtered to 15Å (Fig. 4).

c) The main argument for Pdc4-CTD binding close to the mRNA entry is the direct contact with the well-resolved Pdc4-NTD (up to residue 145 at the mRNA entry); in absence of additional data it remains unclear, if the density provides the correct location of Pdc4-CTD. Therefore, the authors should tone down their statements accordingly, e.g. in lines 172-173 “two MA3 domains” should be replaced with “additional density”, and in line 309 “as observed in our structure” with “as suggested by our structure”.

In the revised version of the manuscript, we have used blush regularization to significantly improve the quality of the map. We can now utilize the crystal structure of Pdc4-eIF4A for rigid-body fitting, facilitating the identification of eIF4A and Pdc4-CTD within the structure.

4. In the methods it should be clarified, whether 43S/48S complexes for native gel assays were assembled in presence of Pdc4, or if Pdc4 was added after 43S/48S assembly. In this line, it would be interesting to know (this is no request!), if the authors tried pre-incubation of Pdc4, or Pdc4-NTD with vacant 40S subunit before adding eIFs and tRNA to increase binding to the 43S complex, independent of eIF4A?

Thank you for this insightful comment. We have tested 40S preincubation with Pdc4 prior to adding eIFs or the other way around and observed no difference in a native gel analysis. We believe incubation of the final mixture at 37 degrees for 10 min is sufficient to reach equilibrium, where the order of addition or preincubation with some components does not make any difference. This is further quantitatively verified in an anisotropy competition assay (Fig. 1B), where we see no change in the inhibition constant when we added the competitor to the 40S-FL-Pdc4-NTD complex preincubated at 37 degrees for 10

min compared to no preincubation. This indicates Pdc4 binding kinetics is fast enough to have multiple turnovers to reach equilibrium during incubation. We have clarified this point in Methods as suggested.

5. a) In Figure 5A/B co-migration of Pdc4 with 43S complexes w/o eIF4A, and of Pdc4-NTD with 43S complexes in general, is hard to recognize. Can the authors exclude, particularly for the full-length Pdc4, that this results from a smearing of the other additional bands observed under these conditions? The corresponding legend should acknowledge the differences at least by adding a quantitative statement.

To confirm if the concerned band in lane 3, Fig. 5A is indeed FL-Pdc4 in complex with the 43S, we generated a new native gel where FL-Pdc4 is competed off from the 43S by 30-fold excess unlabeled Pdc4 (new Extended Fig. 9A, done in a manner similar to Extended Fig. 9D but in the absence of eIF4A). The fraction of FL-Pdc4 overlapping with the 43S has almost disappeared by an addition of the competitor, while the free FL-Pdc4 band is enhanced.

Although an accurate quantification is a challenge for such a weak band, we estimate <20% of FL-Pdc4 is bound to 43S in the absence of eIF4A, compared to that bound to the 40S, which is now described in the Fig. 5 legend as suggested.

b) In Figure 5C, the overlap of Pdc4-CTD band with 43S prevents interpretation of 43S binding, which is clearer in Extended Data Fig. 9A. Therefore, a reference to the latter in the Figure 5 legend would be helpful. Moreover, in line 509 using “free eIF4A” instead of “eIF4A on its own” might improve legibility.

Thank you for this suggestion. We modified Fig. 5 legend accordingly. For a similar reason, we also moved some sentences in the original main text (lines 193-196 and 206-209) into the Fig. 5 legend for readability.

6. In Figure 3B, the authors show that Pdc4-NTD might interfere with Pin initiator tRNA binding in the closed 48S initiation complex by a slight steric clash. However, as Pdc4-NTD strongly precludes mRNA binding, and thus 48S formation, the more relevant question is in how far the NTD might interfere with Pout tRNA in human 43S complex, which has no mRNA bound (PDB 7A09). Figure 3B should be replaced accordingly.

Thank you for this valuable suggestion. We agree that a comparison with open/P_{out} 43S is relevant. We originally superposed tRNA^{Met} in the closed/P_{in} conformation because the unique 40S conformation in our structure is much closer to the closed conformation of the ribosome, and thus likely resents how an incoming tRNA might be incompatible when Pdc4 is already on the 40S and induced the head swivel.

We now significantly extended the structural comparisons around the A- and P-site to show that possible clashes of NTD with eIF1A and tRNA are likely dependent on the 40S conformation. Also, we now describe that the stability of the NTD on the 40S appears to be highly dependent on the unique head swivel in the current structure, because the extensive interaction between NTD and the mRNA binding cleft involves many 18S rRNA bases in both head and body. Therefore, opening of the 40S should significantly alter this interaction and the stability of NTD as well as the degree of steric hinderance that may be caused by the NTD.

Accordingly, we made a substantial edit under the subtitle “Pdcd4-NTD blocks the decoding center of the 40S subunit” with a new Fig. 3 including a superposition with the open/P_{out} 43S as reviewer 1 suggested. We believe this is now more consistent with our native gel analysis, which shows stable coexistence of Pdcd4, eIF1A, and eIF2 ternary complex on 43S in the presence of eIF4A.

7. From Extended Data Figure 4 it is not clear in how far Pdcd4-NTD and SERBP1 sequences correlate with their structures on the 40S; this information should be added in the figure.

We have now revised Extended Data Fig. 4C to include the multiple sequence alignment of regions corresponding the mRNA channel binding domain, based on the structural comparison.

Minor points

8. In line 133, to be more specific, the authors should replace “the affinity of an RNA”, with “the affinity of an mRNA”.

We modified the description as suggested.

9. In lines 281-285, the authors might mention their observation of Pdcd4 comigrating with 80S ribosomes.

Thank you for a valuable suggestion. The new Discussion section now includes a brief speculation about Pdcd4 binding to 80S in the sucrose gradient. In addition, we also discuss if Pdcd4 might inhibit eIF4F in addition to the entry channel eIF4A in more detail than in the original lines 281-285.

10. In Figure 7 it would be helpful to show the Pdcd4 schematic always with the same scaling. Moreover, the figure might be supplemented by information on the nutrient-dependence of Pdcd4 localization.

In response to your suggestion and to enhance clarity, Figure 7 has been revised to show Pdcd4 with consistent scaling instead of a 3D perspective. Additionally, “stress” has been included in the figure for comprehensive representation.

11. It would be helpful for the reader to mention in the sorting scheme for the 40S-eIF1-eIF3-Pdcd4 structure (see point 1b) that complexes were obtained in the context of 48S assembly.

The revised Extended Data Fig. 1 now illustrates a comprehensive cryo-EM data processing workflow.

12. In several figure legends references to panels are missing, e.g. for Figure 2A, Figure 4C, Extended Data Figure 3B, Extended Data Fig. 4B.

We thank the reviewer. We have edited the figure legends accordingly.

Reviewer #2 (Remarks to the Author):

The tumor suppressor Pdcd4 is a translation regulator that consists of two N-terminal RRM motifs and the middle and C-terminal MA3 domains. It was shown that the N-terminal RRM motifs are required for

association of Pcdc4 with 40S ribosomal complexes, whereas the MA3 domains bind to eIF4A inhibiting its helicase activity. However, the exact mechanism by which Pcdc4 influences translation has remained unresolved. Here, Querido et al. report the cryo-EM structures of 40S/Pcdc4 and 40S/eIF3/eIF1/Pcdc4 complexes to gain more insights into this process. The first structure revealed the ribosomal position of a small N-terminal fragment (a.a. 99-145) of Pcdc4 bound in the mRNA channel of the 40S subunit. The second structure additionally contained density positioned at the mRNA entrance (interacting with the eIF3bgi module) which was tentatively attributed to the MA3 domains of Pcdc4, even though the resolution was too low to allow modelling of its structure.

Interestingly, the binding site and the structure of the 40S-binding domain of Pcdc4 resembled those of ribosomal hibernation factors Serbp1/Stm1p. The functional importance of ribosome-interacting residues in Pcdc4 was shown using appropriate mutants in 40S/Pcdc4 binding experiments. Importantly, the region of Pcdc4 bound in the mRNA channel would clash with eIF1A and the eIF2/GTP-Met-tRNA^{iMet} ternary complex. However, although the Pcdc4 N-terminal domain (a.a. 1-156) and full-length protein had comparable affinities for the 40S subunit, it did not inhibit translation, suggesting that the MA3 domains that are known to interact with eIF4A are essential for inhibition. On the other hand, the individual C-terminal MA3 domains alone were also not able to induce potent translation inhibition, indicating that the presence of both N- and C-terminal regions of Pcdc4 is required for the Pcdc4 function.

The position of the additional density at the mRNA entrance in the 40S/eIF3/eIF1/Pcdc4 complex that was attributed to the MA3 domains was compatible with their interaction with eIF4A bound at its second ribosome binding site, which is also at the mRNA entrance. This prompted the authors to investigate the influence of eIF4A on association of Pcdc4 with 40S ribosomal complexes. Indeed, using native gel shift assay, the authors were able to demonstrate eIF4A-induced stimulation of binding of full-length Pcdc4 to, what the authors claim, are 43S preinitiation complexes (see Major critique).

In conclusion, this is an interesting study, in which the conclusions derived from the rather incomplete cryo-EM structures are supported by biochemical data.

MAJOR CRITIQUE

Regarding the facts that (i) binding of Pcdc4 in the mRNA-binding channel would cause a clash with eIF1A and eIF2-ternary complex and (ii) that the authors consistently did not find the class in which Pcdc4 is bound to the 40S subunit in the presence of eIF2/Met-tRNA^{iMet} (even though the mixture for 48S complex formation, which yielded the 40S/eIF3/eIF1/Pcdc4 complex, by definition did contain eIF2/Met-tRNA^{iMet}) raise the question of the exact composition of 40S complexes to which Pcdc4 is bound in Figure 5. These could be not the bona fide 43S complexes, but instead 40S/eIF3/eIF1 complexes, exactly those, which the authors observed in cryo-EM. The authors must repeat these experiments using side by side 43S and 40S/eIF3/eIF1 complexes with/without eIF4A.

It is also surprising that the authors do not detect eIF4A associated with the 40S/eIF3/eIF1/Pcdc4 complex, even though ribosomal complexes were stabilized by cross-linking before being applied to the grids. What was the difference in the conditions for complex preparation etc. compared to those used by the same group when they detected eIF4A at the mRNA entrance? In fact, the presumable Pcdc4/eIF4A interaction at that location would mutually stabilize ribosomal binding of both proteins.

Thank you for raising this important point. To confirm whether eIF1A and eIF2-TC are present in the 43S in Fig. 5 native gels, we generated a new native gel to more rigorously compare 40S-eIF1-eIF3 and the 43S as suggested by the reviewer (new Extended Fig. 9C). In the presence of eIF4A, addition of eIF1A and the eIF2 ternary complex to the Pdc4-40S-eIF1-eIF3 complex cause a minor but distinct shift to form the full 43S. In the same gel, FL-Pdc4 clearly comigrates with the complex regardless of eIF1A and eIF2-TC. Thus, we believe that this result demonstrates that eIF1A, the eIF2 ternary complex, and Pdc4 can stably coexist on the 43S in the presence of eIF4A in Fig. 5A, lane 4.

Based on this, we now discuss that either Pdc4-NTD or the 43S (or both) would likely adopt an alternative conformation(s) to co-bind at the A- and P-sites (see section under “Pdc4-NTD blocks the decoding center of the 40S subunit” and Discussion). We further emphasize that Pdc4-NTD interaction with the mRNA entry channel is still crucial for Pdc4 binding to the 43S in the presence of eIF4A (Extended Fig. 9D), suggesting that the critical interactions observed in our structure are also important in the 43S complex.

Importantly, we now describe Pdc4-CTD in complex with eIF4A in a new structure. Given the stable binding of Pdc4 to the bona fide 43S in the presence of eIF4A in solution (new Extended Data Fig. 9C), it seems surprising to see that this structure still lacks eIF1A and the eIF2 ternary complex. It's possible that a minor difference in conditions used to reconstitute the complex for biochemical assays and cryo-EM could cause this difference. The fact that we observe the current structure to be a minor subpopulation (~1%) on the grid may also explain why we are not able to observe what might be a less stable complex on the grid. As discussed above, it is also possible that a conformation of Pdc4 could be more flexible in the 43S/48S complex and thus did not survive through the 3D reconstruction process.

Despite our biochemistry implying that it does not necessarily prevent the formation of the 43S, the specific conformation of Pdc4 in the current structure also implies that inhibition of factor binding to the mRNA binding channel may occur under certain context, a possibility now discussed in the Discussion.

It is noteworthy to mention that complexes consistent with our structures are present in vivo, as evidenced by a recent study of native Pdc4-ribosomal complexes (PMID: 38641729). In this study, even when using affinity purification, a mere 1.9% of the particles corresponded to the Pdc4-40S- eIF4-eIF3-eIF1 complex.

MINOR COMMENTS

1. It is an exaggeration to claim (line 257) that the structure of human Pdc4 has been determined on the ribosome. The authors were able to model the structure of a fragment corresponding to 46 out of a total of 465 a.a. residues.

In the revised version of the manuscript, we have employed blush regularization to enhance the quality of the map. This enhancement allows us to use the crystal structure of Pdc4-eIF4A for rigid-body fitting, which facilitates the identification of eIF4A and Pdc4-CTD within the structure.

2. The authors should explain what an MA3 domain is (line 52).
It would be helpful if Figure 1A showed the borders of the two RRM domains.

In the revised manuscript, we now briefly explain that the MA3 domains are homologous to eIF4A binding domains in eIF4G, and binds eIF4A to inhibit its activity. We also modified Fig. 1A as suggested.

3. The authors should comment on their finding that the Pdc4-CTD does not inhibit translation in light of reports that even the C-terminal MA3 domain alone inhibits translation in transfected cells (e.g. Laronde-Leblanc et al, MCB 2007 (PMID: 17060447)).

We also observe that the Pdc4-CTD fragment can inhibit lysate translation at a high concentration, suggesting a possibility that Pdc4 functions through multiple pathways, for example by inhibiting eIF4F outside ribosomes. We now included this in Discussion with citing the suggested paper.

4. The title page shows that the institutional affiliation of M. Sokabe and CS Fraser is with the University of Michigan; although UC Davis is listed, none of the authors is affiliated with it.

Thank you for pointing this out. We corrected it accordingly.

5. The Methods section has been carelessly prepared, and contains numerous typographical errors, grammatical errors and omissions.

Line 464. The method may be “widely described”, but the authors should nevertheless cite an appropriate reference

Line 465, 502. The quality of English throughout Methods should be improved e.g. “after harvested and washed” (line 465) and “were measure in the nanodrop’ (line 502).

Line 473. What is the “translation reaction buffer”?

Line 475. Correct “b-globin 5c UTR”

Line 484. What is “PFA”?

Lines 486-7, 510. The sources of antibodies should be mentioned.

Line 498. ‘trough’ should be ‘through’

Line 516. Missing reference.

Lines 521-522. Missing reference.

Line 547. The PMID number is not necessary here.

Lines 563, 567. It seems unlikely that the reaction volume was 25ml.

Line 581. The cited reference does not refer to Relion4.

We thank the reviewer for these comments, and modified them accordingly.

Reviewer #3 (Remarks to the Author):

Pdc4 inhibits mRNA translation in response to stress, although the molecular mechanism is not entirely understood. This manuscript takes a structural approach to understanding how Pdc4 binds to and inhibits the ribosome. The authors show that the N-terminal domain of Pdc4 inserts into the mRNA entry site of the 40S subunit. This likely prevents mRNA binding, given that it would sterically clash with

Pdcd4. In addition, this anchors Pdcd4 to the 40S, allowing the C-terminal domain of Pdcd4 to interact with eIF4A and to act.

Overall, I think this manuscript provides useful and interesting insights into the molecular mechanism how Pdcd4 can function. It will be of interest to a wide audience of people studying mRNA translation and stress responses. I found the data solid and convincing. I have only very minor comments:

Minor:

1. Fig 4B-C: it's not clear which is the cryoEM density (grey?) and which is the rigid-body fitting of the known crystal structure of Pdcd4-CTD (pink?).

It would be good to label them.

In the revised manuscript, we have employed blush regularization to enhance the quality of the map. Figure 4 has been substituted with a new figure showing the architecture of Pdcd4-40S-eIF4A-eIF3-eIF1.

2. Fig 5, the labeling of lane 1 ("WT") is confusing because for all the other lanes the label describes what is present in addition to Pdcd4, so it should actually say "-". Otherwise it seems like WT ribosomes are added? (Same for 5B-C).

Thank you for pointing this out. We agree that labels in the original figure was confusing, and now modified Fig. 5A-C accordingly.

REVIEWERS' COMMENTS

Reviewer #1 (Remarks to the Author):

The authors appropriately addressed all my points. However, there are some points in the revised version, which should be edited.

1. The authors now describe a “key interaction” between eIF3g-RRM and Pdcd4, forming an intermolecular β -sheet (lines 192-195, 210-212 and 304-305). From the corresponding Figures 4 and 7 it is not clear, if the experimental density allows reliable modeling of this interaction. The authors should clarify on what basis they modeled the interaction.
2. Extended Data Fig. 1, bottom panel: “40S-Pdcd4” should be replaced with “48S-Pdcd4”.
3. Extended Data Fig. 2: In E) and F) the unit Å is missing, in E) the X-axis label 0.5 seems incorrect.

Reviewer #2 (Remarks to the Author):

Brito Querido et al. have substantially revised their manuscript, both in response to my previous comments, which have been addressed satisfactorily, and by taking advantage of technical advances to improve the local resolution of their cryoEM analysis, which has enabled them to identify and assign additional density. The manuscript makes valuable contributions to understanding translational control in humans.

REVIEWERS' COMMENTS

Reviewer #1 (Remarks to the Author):

The authors appropriately addressed all my points. However, there are some points in the revised version, which should be edited.

1. The authors now describe a “key interaction” between eIF3g-RRM and Pdcd4, forming an intermolecular β -sheet (lines 192-195, 210-212 and 304-305). From the corresponding Figures 4 and 7 it is not clear, if the experimental density allows reliable modeling of this interaction. The authors should clarify on what basis they modeled the interaction.

In the revised manuscript, we described that this interpretation was based on secondary structure prediction and cryo-eM map.

2. Extended Data Fig. 1, bottom panel: “40S-Pdcd4” should be replaced with “48S-Pdcd4”.

In the revised Extended Data Fig. 1, we have replaced “40S-Pdcd4” with “48S-Pdcd4”.

3. Extended Data Fig. 2: In E) and F) the unit Å is missing, in E) the X-axis label 0.5 seems incorrect.

We appreciate the reviewer's insightful feedback. The figure has been revised accordingly to reflect the suggested changes.

Reviewer #2 (Remarks to the Author):

Brito Querido et al. have substantially revised their manuscript, both in response to my previous comments, which have been addressed satisfactorily, and by taking advantage of technical advances to improve the local resolution of their cryoEM analysis, which has enabled them to identify and assign additional density. The manuscript makes valuable contributions to understanding translational control in humans.